

# Does predictability of fluxes vary between FLUXNET sites?

Ned Haughton[1], Gab Abramowitz[1], Martin G. De Kauwe[1], and Andy J. Pitman[1]

[1]Climate Change Research Centre, UNSW Australia

*Correspondence to:* Ned Haughton (ned@nedhaughton.com)

**Abstract.** The FLUXNET dataset contains eddy covariance measurements from across the globe, and represents an invaluable estimate of the fluxes of energy, water and carbon between the land surface and the atmosphere. While there is an expectation that the broad range of site characteristics in FLUXNET result in a diversity of flux behaviour, there has been little exploration of how predictable site behaviour is across the network. Aside from intrinsic interest in this fundamental question, under-standing site predictability would be useful for land surface model (LSM) evaluation in setting a priori expectations of model performance. It would also provide a clear rationale for selecting particular FLUXNET sites for model development, evaluation and benchmarking. Here, 155 datasets with 30 minute temporal resolution from the Tier 1 of FLUXNET2015 were analysed in a first attempt to assess individual site predictability. Predictability was defined using the disparity between the ability to simulate fluxes at a site given specific knowledge of the site, and the ability to simulate fluxes given general land surface spec-ifications. We then examined predictability using performance metrics including RMSE, correlation, and probability density overlap, and defined site **uniqueness** as the disparity between multiple empirical models trained globally and locally for each site. A number of hypotheses potentially explaining site predictability were then tested, including climatology, data quality and site characteristics. We found very few clear predictors of uniqueness across different sites including little evidence that flux behaviour is well discretised by vegetation types. While this result might relate to our definition of uniqueness, we argue that our approach is sound and provides a useful basis for site selection in LSM evaluation.

## 1 Introduction

The land surface is a key component of the climate system, as it provides feedbacks to atmospheric conditions via the exchange of heat, moisture, and carbon fluxes. These surface-atmosphere exchanges are contingent on the characteristics of the soil and vegetation. However, these interactions between the atmosphere and land are not uniform, for example in hot, mesic environments net primary productivity (vegetation productivity) becomes less sensitive to the effect of climate (Bonan, 2015; e.g. increasing precipitation Chapin III et al., 2011; Del Grosso et al., 2008; Gillman et al., 2015; Huston and Wolverton, 2009; Schuur, 2003). Across the globe, variability in the productivity-climate relationship suggests that the behaviour of some ecosystems must be more predictable than others. Intuitively, the behaviour of ecosystems that experience marked stochasticity in precipitation (e.g. ecosystems that rely on monsoonal rains for growth), would likely be harder to predict than ecosystems that experience relatively consistent year-to-year conditions (e.g., the boreal zone, the wet tropics or desert regions). Furthermore, whilst vegetation patterns are broadly understood at global scales (e.g. the Köppen climate classification Kottek et al., 2006;



and Whittaker's biome classification, Whittaker, 1962), at local and regional scales, plants exhibit local scale adaptations to their environment (e.g. soils and topography). Taken together, these relationships between climate and local factors point to a spectrum of site predictability. Perhaps surprisingly, the predictability of a site is rarely considered when choosing sites to evaluate models.

Within the climate science community, 30-years of research originating from Dickinson et al. (1986) has transformed how land surface models (LSMs) describe the exchange of energy, water and, more recently, carbon. The community has moved from no explicit representation of vegetation in climate models (e.g. Manabe, 1969), through highly empirical second generation models (which included an explicit simulation of heat and moisture fluxes and storage in soil and vegetation), to models that attempt to explicitly represent the soil-vegetation continuum (see review by Pitman, 2003). Common to virtually all LSMs
is an assumption that flux behaviour variations between biomes, given similar driving conditions, can be explained by a small sample of structural and physiological parameters, grouped as plant functional types (PFTs). As a result, land modellers have sought observations from locations characteristic of these broad PFTs to develop and evaluate models.

Land surface modellers often use FLUXNET data to evaluate their models, and to tease out weaknesses, with the goal of model improvement. In the 1990s and 2000s, when only a handful of flux tower site measurements were available, the
LSM community gravitated to these, hence observations taken at Cabauw (The Netherlands), Harvard Forest (USA), and near Manaus (Brazil) were widely used. Over the last two decades, direct measurement of land surface fluxes and meteorological variables has rapidly expanded, as new flux towers are installed and existing towers continue to gather data. FLUXNET 2015 (Fluxdata.org, 2018), as a synthesis of these measurements, represents a rich source of information about the exchange of carbon, water and energy. The freely available release (Tier 1), encompasses over 150 sites and includes over 500 site-years of
high temporal, quality-controlled data. These data provide an unparalleled opportunity to improve our observationally-based understanding of land-atmosphere exchanges of carbon, water and energy. They are also particularly useful for LSM evaluation since both the necessary driving variables (meteorological variables) and prediction variables (energy, water and carbon fluxes) are reported at a spatio-temporal scale relevant to LSMs. As a consequence, land surface modellers have developed tools to enable the FLUXNET 2015 data to be used routinely (e.g. Ukkola et al., 2017). However, with hundreds of site datasets now
freely available, site choice for model evaluation varies widely among the land surface community, with no common strategies for site selection. FLUXNET sites differ in many ways: in data record length (from less than one to greater than twenty years); in climate regime; and in soil and vegetation characteristics. Their similarity to each other also varies – FLUXNET is not evenly distributed over the globe, and has higher density in more densely populated and wealthy regions, such as Western Europe, and the north-east of the United States, with particularly heavy representation of temperate forests.

Despite obvious distinctions between sites in FLUXNET defined by precipitation regime, temperature, seasonal snow cover and indeed PFT type, it is not immediately clear which of the 150-plus freely available sites are most useful for model evaluation. One might assume that given the diversity of sites, some are easier to simulate than others, and it seems sensible to assume that the choice of sites could have an impact on insight gained from model evaluation at these sites. However, assumptions about the predictability of different sites have not been explicitly tested. For example, in recent multi-model evaluation and
benchmarking experiments, where multiple FLUXNET sites were used, Best et al. (2015), Haughton et al. (2016) and Haughton





et al. (2018) were not able to identify any obvious patterns in model performance across sites. The lack of quantification of predictability means that site selection for evaluation is potentially susceptible to confirmation bias. That is, a modeller might unconsciously choose sites that are easier for their model to simulate, rather than selecting sites based on their instructiveness for identifying flaws in a model. For example, consider the implications of evaluating a model against ten FLUXNET sites

that happen to be the least predictable in comparison to evaluation against the ten most predictable sites. In the former case, a modeller might become disillusioned with the apparent lack of skill of a potentially good model, while in the latter case a modeller might become overconfident concerning the skill of a poor model.

This issue of site predictability has been ignored in historical flux-model comparisons, where modelling groups have generally not tried to justify their choice of sites, or based their reasoning around issues such as data availability or length of record.

Chen et al. (1997) chose the Cabauw site for a multi model intercomparison because it was considered relatively easy to simulate. Several authors chose longer (multi-year) sites (Balsamo et al., 2009; Lawrence et al., 2011; Wang et al., 2011). Some evaluation papers explicitly sought to sample a range of PFTs (Bonan et al., 2014; De Kauwe et al., 2015). Many highlighted choices based on the availability of gap-filled data (Krinner et al., 2005; Slevin et al., 2015; Wang et al., 2011). A few papers highlighted the high natural variability of a site (Balsamo et al., 2009), or a high degree of climate differences between sites

(Wang et al., 2011). Others highlighted the quality of specific sites and some provided evidence for this decision based on energy closure (Napoly et al., 2017). In contrast to the often detailed explanation for why a specific model or parameterisation is chosen, the defence of specific evaluation data sets often lacks a coherent rationale. Most commonly, "high quality" or "longer" data sets are selected. A longer data set may sample more years, but a single month of data from another site might provide more information regarding a specific phenomenon (e.g. the response to a drought or a heatwave). Sampling more PFTs might

be valuable, but might also bias results if the selected sites fall within a similar behavioural regime not well discretised by PFT. In short, it would be useful to be able to make clear, evidence-based statements about the relative predictability of different sites, based on meteorological patterns or local site characteristics. This would allow modellers to make informed site selection choices for model development/evaluation that maximise coverage of diverse site behaviours, and ultimately help to reduce uncertainty in model projections.

There is no single definition of predictability, but it can broadly be defined as the ability to reproduce a property of a system, given only knowledge of variables that are causally related to that property. Predictability of a system should therefore also encompass the capacity to predict changes in the property of interest, given changes in the drivers of the system, for example differing flux responses in wet and dry periods. In this context we might envisage predictability to be the degree to which a "perfect" model could accurately estimate measured fluxes at a site, given appropriate meteorological variables and relevant

site characteristic information. Of course, we lack a perfect model, and the accuracy of our observational data is always limited by measurement error, and noise in the system being measured. As such, any practical measure of predictability will be limited in accuracy, but this does not mean that it cannot still be useful.

Some predictability metrics do exist: Colwell (1974) defines a predictability metric based on constancy in time and contingency on season but this metric only captures one aspect of performance – temporal correlation. Abbas and Arif (2006) also

proposed a number of time series predictability metrics, but these are only useful in univariate time-series prediction, where the





forecast is made only given knowledge about the predicted variable itself, rather than knowledge of other predictor variables, as is the case with flux prediction from meteorological variables. There are also model-class specific performance metrics, such as the Genetic Programming predictability metric presented in Kaboudan (2000), but such metrics rely on the assumption that the model is suitable for predicting the data in question.

Since existing predictability metrics are not suitable to our problem, below we detail a new metric of site predictability and analyse the FLUXNET 2015 sites according to their predictability. To do this, we applied a suite of empirical models to predict fluxes at the 155 flux tower sites with half-hourly data included in the Tier 1 FLUXNET 2015 release. We also investigate several hypotheses that might explain the variation in site predictability in different locations. Finally we attempt to provide a sound theoretical basis for site selection for LSM development and model intercomparison projects. This will allow *a priori*

expectations of model performance to be better defined, as well as mitigate the potential for ad-hoc site selection to shape judgement of how well LSMs perform.

## 2   Methods

Differences in predictability between sites might be due to many factors, including, but not limited to:

- variability of meteorology (e.g. strong seasonality in precipitation compared with low variability, large seasonal cycles
in incoming radiation compared to small seasonal cycles, and stochastic events);

- complexity or consistency of the site itself (e.g. orographic effects, managed land use including different irrigation and cropping patterns, vegetation and soil structures);

- broader scale impacts (e.g. climate type, regional aridity, teleconnections to major oceanic drivers, landscape hetero-geneity, geological basins);

- technically sourced variance (quality of instrumentation, assumptions and application of eddy covariance methodology, post-processing).

We focus on the first of these and ask whether predictability at a specific site can be understood in terms of the differences in flux behaviour given particular site and meteorological conditions, relative to the flux behaviour that would be expected at other sites given the same conditions. We do this by training a suite of empirical models (based on the models described in Haughton

et al., 2018) to predict fluxes, based on meteorology, at each FLUXNET site twice. First we train the empirical models using all of the available data from all of the available sites at once ("global training"), to characterise the general expected flux behaviour given a specific set of meteorological conditions. Then we re-train the models using only data from the individual site in question ("local training"). The globally and locally trained versions of the models are then used to make predictions at each FLUXNET site, and their performances are compared, using a range of performance metrics. Any improvement in

performance by the locally trained model over the globally trained model is an indication of driver-flux relationships that are unique to the site in question (note that this may include systematic errors in measurement). Since such a site exhibits



relationships between drivers and fluxes that are not broadly shown at other sites, we argue this site has lower predictability than a site that acts more similarly to the global behaviour.

To quantify this, we plot the local and global metric values as Cartesian coordinates, then convert them to polar coordinates (see Figure 1). The origin represents the best possible performance metric value, so distance to the origin represents the mean

site performance across the global and local simulations. The degree to which each point drops below the 1:1 line will be our definition of uniqueness, or lack of predictability. To illustrate, imagine a model that perfectly represented all relevant process and fully utilised all of the available information in the input data to make the best possible prediction. This model could be used to assess site predictability based on the residual sum of squares against observations, and this metric value could be compared across different sites. No such model exists of course, and we therefore use empirical models to assess the

predictability of the data while minimising assumptions about the functional form of any relationships between variables. For further discussion of why empirical models are suitable for estimating the information available in FLUXNET data, see Best et al. (2015) and Haughton et al. (2016).

In particular, we have used models in the framework developed in Best et al. (2015) and Haughton et al. (2018), to predict net ecosystem exchange (NEE), sensible heat (Qh), and latent heat (Qle). These models included some simple linear regressions,

as well as cluster-plus-regression models (K-means clustering over meteorological driving data, and then an independent linear regression between drivers and fluxes at each cluster). Models used various combinations of meteorological driving variables: down-welling shortwave radiation (S), surface air temperature (T), relative humidity (H), wind speed (W), and precipitation (P). Models also used a number of derived variables, including difference in air temperature and specific humidity since dawn (dT and dQ, respectively), as well as lagged averages of each meteorological forcing (e.g. lH10d indicates a 10-day lagged average

of H), where the lags were chosen pseudo-optimally. Haughton et al. (2018) showed that each of these driving variables we use here added predictive power to the models, and had relatively low correlation, avoiding problems of collinearity. Models referenced below follow a standard naming scheme that indicates the structure of the model, for example, *S_lin* indicates a linear regression using only shortwave down, while *STHdT_lS30d_km243* would indicate a cluster-plus-regression model with 243 clusters, with shortwave down, air temperature, relative humidity, temperature difference since dawn, and a 30-day lagged

average of shortwave down as inputs. None of the models are provided with site characteristic data (e.g. geographic, soil, or vegetation information) as we want to use the models to test the effects of these characteristics on predictability. A complete list of the empirical models is shown in Table 1.

To run the models, we converted the raw FLUXNET 2015 Tier 1 data (only sites with half-hourly data, 155 in total), using the FluxnetLSM tool developed by Ukkola et al. (2017). In all cases, the empirical models are trained only on high-quality

non-gap-filled data, according to Quality Control (QC) flags from FLUXNET 2015 and FluxnetLSM. The models are then run on all available data (including gap-filled data, to maximise the time coverage of empirical models with time-lagged drivers), and evaluated only on time steps with non-gap-filled data. We then plotted each metric value for each site in a scatter plot, with the global value on the x-axis, and the local value on the y-axis. We decomposed that information into:





| model name | Successful simulations | | | | Negative uniqueness | | |
|---|---|---|---|---|---|---|---|
| | global | NEE | Qh | Qle | rmse | corr | overlap |
| S_lin | 155 | 155 | 155 | 155 | 2 | 206 | 120 |
| ST_lin | 155 | 155 | 155 | 155 | 1 | 20 | 117 |
| STH_km27 | 152 | 152 | 152 | 152 | 3 | 4 | 50 |
| STH_km81 | 152 | 146 | 150 | 149 | 3 | 4 | 39 |
| STH_km243 | 152 | 108 | 133 | 131 | 3 | 3 | 25 |
| STHW_km81 | 152 | 142 | 148 | 147 | 2 | 3 | 23 |
| STHW_km243 | 152 | 88 | 126 | 123 | 2 | 2 | 11 |
| short_term243 | 152 | 65 | 98 | 97 | 1 | 1 | 9 |
| long_term243 | 152 | 3 | 12 | 12 | 2 | 2 | 0 |
| long_term729 | 152 | 0 | 0 | 0 | 0 | 0 | 0 |

**Table 1.** number of sites that models ran successfully at, for global training (columns 1), the number successful local simulations for each variable (columns 2-4), and the number of cases of negative uniqueness, indicating that the local model performed worse than the global model, for each of the three metrics (columns 5-7). Three sites (CA-Man, DE-RuR, and DE-RuS) did not include relative humidity, and so all models including that variable failed, including the global model.

1. Mean performance: the mean metric value over the local and global simulation at each site, defined by distance from the origin. Higher is worse in most metrics, including root mean square error (RMSE), but lower is worse in the case of Pearson's correlation coefficient (Corr) and Perkins' distribution overlap metric (Overlap, Perkins et al., 2007).

2. "Uniqueness": the angle below the 1:1 line. Uniqueness is calculated as $\frac{4}{\pi}arctan(\frac{x-y}{x+y})$, such that if, for example, RMSE is 0 locally and some positive value globally, uniqueness will be 1.

Note that because the best possible result for some metrics is 1 (e.g. Corr and Overlap), in those cases we subtract the value from 1 such that the best result is 0 before calculating the uniqueness, so that it can be interpreted the same way across metrics - that positive numbers indicate better local performance. We avoid transforming metrics for mean performance, so that metrics are in their standard units.

In general, this definition of uniqueness ranges from -2 to 2, and is strictly between -1 and 1 for metrics that only have values on one side of "best" (e.g. RMSE is positive definite, Corr is always less than or equal to 1), but in most cases should lie between 0 and 1. A model's uniqueness is 0 if the local and global simulations perform equally well, between 0 and 1 if the local model performs better than the global model, and negative if the local model performs worse that the global. Negative values are unusual, and indicate that the local meteorological forcing provides insufficient useful information to increase performance, and that the local model has failed in a spurious way (discussed in more detail below). The number of negative uniqueness values for each metric and each model is shown in the last three columns of Table 1, out of a maximum of 10 x 155 = 1550 cases.




The uniqueness and mean performance metrics are shown for RMSE in Figure 1 for the S_lin empirical model to illustrate how to interpret later figures: *uniqueness* is the angle measured clockwise from the origin (the optimal metric value) and the 1:1 line (equal local and global performance), and *mean performance* is the average performance of the local and global simulations, given by the distance of each point from the origin. Each point is a different site. Figure 1 also illustrates the

differences in a simple case between the results when the local training data is identical to the testing data, and when it differs due to mismatch between the meteorological and flux QC flags between training and testing. Both rows of figures show local versus global values, but the first row uses only flux variable QC flags for evaluation, and so can include part of the period for which there are poor QC meteorology values. The second row uses exactly the same QC flags (and so time steps) in both training and testing phases (as a proof of concept). As a result, in the second row, the points are strictly at or below the 1:1 line.

In the first row, the points shift slightly, and some lie very slightly above the 1:1 line. This difference can be exacerbated for more complex models.

Best et al. (2015) used the concept of ranking over multiple performance metrics, and then aggregating over rankings to arrive at a single value that represented a broad concept of performance for each model. This methodology is extremely useful for model evaluation using FLUXNET site datasets. However, due to the very different distributions of results for the different

metrics (discussed below), we avoided aggregating over metrics and instead examined a set of key metrics separately for their ability to capture independent aspects of performance. The metrics we chose were RMSE, as it provides an overview of model accuracy in relevant units, Pearson correlation (Corr) as a measure of temporal correlation, and Perkins' distribution overlap metric (Overlap), as it gives a measure of the match between the observed and modelled distributions.

## 2.1 Caveats

In an idealised experiment, even if we exclude the possibility of over-fitting, the locally-trained model should *always* perform better than the globally trained model to some degree. This is because the local model is predicting the same data that it is trained on, and should capture any behaviour that is site-specific (that is, it is being tested in-sample). However, there are a number of factors that might prevent this from happening.

First, a model may require a substantial amount of data to avoid over-fitting, and some sites may not provide enough data

to train the model locally. For example, very few sites had enough data to adequately train the long_term243 or long_term729 models from Haughton et al. (2018), each of which have 10 input variables (S, T, H, W, dT, dQ, lS30d, lP30d, lH10d, lT6hM) and hundreds of clusters. As such, these models would potentially require hundreds of non-gap-filled data samples at each cluster to obtain a reliable linear regression estimate (so ~$10^4$ samples in total). These more complex models often fail to run locally, or run successfully but produce erroneous results (e.g. due to too few samples to obtain reliable regression results for

a K-means cluster - this problem is described in detail in the Supplementary Material in Haughton et al., 2018). To mitigate this problem, we modified the models from Haughton et al. (2018) to ensure that each cluster always contained a number of samples at least 5 times greater than the number of input variables. When clustering failed, it was re-attempted a further 9 times, and if that was not successful, the model was excluded. See Table 1 for details on how many models ran successfully for each variable.





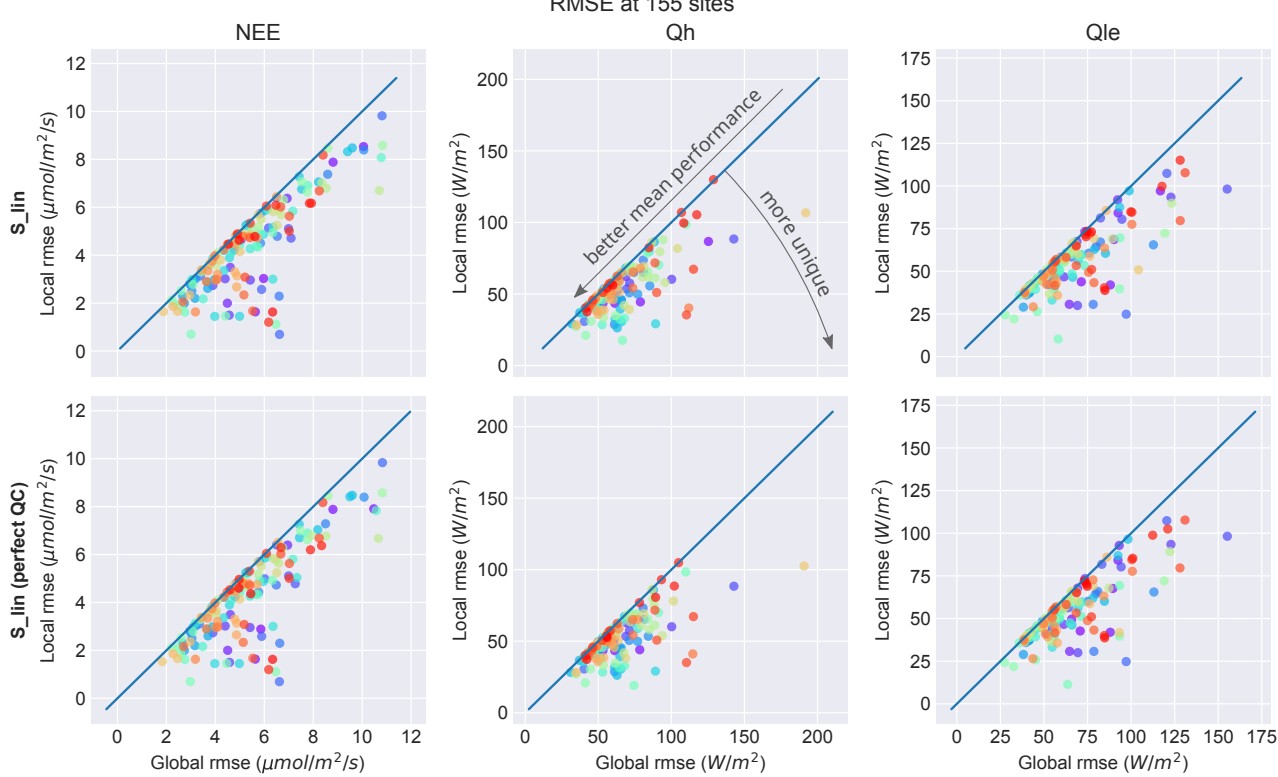

**Figure 1.** RMSE values for the global (x-axis) and local (y-axis) model simulations. Columns show the three fluxes. The top row uses S and the flux QC flags for training, and only the flux QC flags for evaluation (the method used throughout the paper). In the second row, the meteorological QC flags are also used for evaluation, such that the training and testing data are identical. Colours simply serve to identify sites, and allow clearer comparison between the top and bottom rows, and the blue line is a 1:1 line.

Second, as noted above, the training and testing data for the local model are in practice nearly always different, because the QC flags for the flux variables being evaluated against do not correspond perfectly with the meteorological forcing variable QC flags. Models are trained only on data that has good meteorological and flux QC flags for all relevant variables. However, simulations are evaluated on a larger subset of all data – those time steps marked as good QC for the flux variable alone. The motivation for doing this is to ensures that all of the different empirical models are evaluated on the same number of time steps. So, for instance, with the S_lin model predicting Qle at a particular site, the number of time steps with good S *and* good Qle QC flags might be only 80% of the time steps with good Qle QC flags only. Consequently, the model will be trained only on the 80% of period that it is tested on. This problem is exacerbated for models with more inputs and for models with lagged average inputs, which will usually be trained on substantially smaller subsets of data than they are evaluated on.





Lastly, "performance" is dependent on metric, and so performance will only be strictly better locally for metrics that are optimised by the regression-based structure used in the empirical models. For instance, ordinary least squares linear regression optimises RMSE in the training dataset, so assuming the training and evaluation datasets are identical, then the RMSE of the local model will be strictly not worse than the RMSE of the global model. However, metrics which assess model performance

in terms of distribution, such as the distribution overlap metric or temporal correlation, may occasionally show that the local model performs worse than the global model, even when the local model is clearly better under RMSE. This is particularly pertinent in the context of a generally useful predictability metric.

These caveats are worth keeping in mind, but in the majority of the results below, they do not play a particularly large role. We are confident that our predictability metrics are satisfactory for a first attempt to estimate site predictability.

## 2.2 Hypothesis testing

Once we have a predictability metric, we can generate a number of hypotheses about what might determine predictability at different sites. Below we list several hypotheses, many of which intersect, and so in some of these cases we also mapped some predictability metrics against two hypothetical predictability sources.

**Mean annual temperature and precipitation** Sites with higher mean temperature tend to be those closer to the equator, and

tend to have a smaller annual temperature cycle range. All other things being equal, we might therefore expect warmer sites to be more consistent over time, have a more constant response to meteorological forcing, and therefore be more predictable. Sites with higher average precipitation would be expected to have fewer drier periods, more consistently available soil moisture, and higher humidity resulting in a damped daily temperature cycle, and are therefore likely to be more predictable.

For these hypotheses, the FLUXNET site data is not always adequate, as the mean may not be perfectly representative of

the true climatology of the site. For example, if the site only has a short dataset measured over a particularly wet or dry period, or if a site has a strong seasonal pattern in the quality of the temperature data, this would introduce a bias. For this reason, we calculated mean annual temperature and precipitation from the half-degree CRU TS4.01 data (Harris and Jones, 2017), using data from 1961-2016, and using the nearest neighbouring grid cell.

**Aridity** Arid sites tend to have higher precipitation variability, with fewer, heavier rain events, and longer dry periods (Donat

et al., 2016). We would expect that flux predictability would be lower at arid sites. For this hypothesis, we used an aridity index based on mean annual precipitation from CRU TS4.01, and the energy-only estimate for potential evapotranspiration (PET) from Milly and Dunne (2016), based on net radiation and ground heat flux ($PET = 0.8(\overline{Rnet} - \overline{Qg})$) from FLUXNET, such that the aridity index (AI) = mean precipitation/PET. We assumed Qg=0 where sites did not provide Qg (which is approximately true on long time scales).

**Budyko curve deviations** The Budyko curve (Gerrits et al., 2009) plots an evaporative index against a dryness index, with the expectation that sites should, in the long term, fall along a function of dryness that is both energy and water limited. Sites that fall further from the Budyko curve may indicate data errors, or hydrological uniqueness (for example, rapid drainage, or external water sources), or that the data in question is not long enough to adequately capture and account for long-term internal





variability. Whatever the cause of divergence from the Budyko expectation, we would expect that more divergent sites would be more difficult to predict.

**Interannual variability** Sites heavily influenced by longer term climate patterns, such as decadal scale ocean oscillations, are less likely to have all of their relevant patterns captured within the period of FLUXNET measurement, and so potentially

contain systematic biases. We compared the interannual variability between sites for both T and P, using the CRU TS4.01 data. We calculated the coefficient of variance (CoV) for annual means of temperature (K), and precipitation (mm/year). We would expect that as IAV increases (shown by greater CoV), predictability would decline.

**Diurnal ranges** Sites with large diurnal ranges have stronger rates of change between daily peaks and troughs, and these are likely to make prediction harder. Faster changes in temperature, for example, can cause rapid changes in relative humidity,

which is a major driver of latent heat flux. We used the BioClim (WorldClim, 2016) mean diurnal temperature range using the nearest neighbouring grid cell for each site.

**Seasonality** Larger differences between winter and summer conditions would likely lead to lower predictability, since we would expect flux behaviour at such sites to be more diverse over the course of the year. This would also affect the relative influence of time varying factors, e.g. timing of snow melt, or vegetation phenology. For model and site combinations where

the training and testing data is more disjointed, this might also lead to lower predictability due to the non-training testing data diverging more in behaviour. Since about 55% of sites in Tier 1 are less than 5 years long, we used the BioClim variables (WorldClim, 2016) to compare seasonality between sites. We investigated: isothermality - the ratio of diurnal temperature range to annual temperature range; temperature seasonality - the standard deviation of monthly average temperatures, normalised by the annual average in K; temperature annual range; precipitation seasonality; precipitation of wettest quarter; and precipitation

of the driest quarter.

**Vegetation type** The FLUXNET 2015 sites are categorised by International Geosphere-Biosphere Programme (IGBP) vegetation types. There is a widely held assumption that different vegetation types behave differently in response to similar meteorological forcings (although this was assumption was questioned by Alton, 2011), and this presumably also applies to the overall predictability of a site. We grouped IGBP vegetation types into 5 major groups:

– **Evergreen Forest**: Evergreen Broadleaf Forests, Evergreen Needleleaf Forests (49 sites).

     – **Deciduous Forest**: Deciduous Broadleaf Forests, Deciduous Needleleaf Forests (16 sites).

     – **Mixed Forest**: Mixed Forests (7 sites).

     – **Crop**: Cropland/Natural Vegetation Mosaics, Croplands (15 sites).

     – **Grass**: Grasslands (29 sites).

– **Shrubland**: Barren or Sparsely Vegetated, Closed Shrublands, Open Shrublands (11 sites)

     – **Savanna**: Savannas, and Woody Savannas (13 sites).

     – **Wetland**: Permanent Wetlands (15 sites).





Other IGBP vegetation types not represented in FLUXNET 2015 Tier 1 included Snow and Ice, Unclassified, Urban and Built-Up Lands, and Water Bodies. We then compared the performance metrics across these groups.

**Geographic remoteness** We are training the global models on all available sites, but FLUXNET sites are not evenly distributed over the land area of the globe. As such, we might expect that sites that have many other similar sites in the global dataset would have their behaviours more adequately captured by a globally trained model. To investigate whether more geographically unique sites were less predictable, we mapped the sites by uniqueness, and also compared uniqueness by average distance to all other sites.

**Energy balance closure** Wilson et al. (2002) showed that FLUXNET sites often have a problem closing their energy balance. Net incoming radiation (Rnet) does not match the total energy accounted for by the heat fluxes (Qh, Qle, and Qg) and changes in heat storage, on average having an imbalance of around -20% at each site, but ranging from -60% to +20%. Since this imbalance pertains to boundary conditions, which are all measured (sometimes with the exception of Qg, although that can be assumed to be too small to account for the difference on a long enough time scale), the imbalance indicates some problem with either the measurement system, or the eddy covariance methodology. We would assume that sites with worse energy imbalances are likely to be more difficult to predict. We calculated the energy closure gap as the energy_gap = mean(Rnet - Qh - Qle - Qg) (we used Qg=0 for sites missing Qg), and also compared sites by normalised energy gap, using abs(1 - energy_gap/Rnet). Note that this is not the exact formulation used by Wilson et al. (2002), but it serves the same purpose – to identify energy closure imbalances.

**Record length** Since many of the longer-term or rarer behaviours mentioned above are more likely to be captured adequately in site datasets that span longer periods, we should expect that longer sites would be more predictable. On top of this, site principal investigators are likely to become more familiar with problems with their sites, equipment, or methods, and more likely to be able to find solutions to those problems over time, and so we should expect that data quality should improve in longer site datasets. We examined the number of years in the dataset as a predictor for uniqueness.

**Gap-filling ratio** Some bad data is likely to make it through quality assurance procedures, and such bad data would make prediction more difficult. It is not clear how one would tell such data in most cases, unless patterns are obvious. We visually inspected the time series plots produced by FluxnetLSM for each relevant variable, for each site, and saw no obvious problems within the data periods marked as good QC. However, some proxy for data quality may be possible, and in particular sites with more high quality data may indicate better instrumentation or procedures, and less likelihood of having bad data marked as high quality. We compared sites by the proportion of data marked as good QC to total data, averaged over all variables, separately for meteorological and flux variables.

We note that some determinants of predictability could not be calculated for some sites. For example, a number of sites have no non-gap-filled data for precipitation, and so mean annual precipitation can not be calculated, and neither can dependent determinants, such as aridity index. In such cases, the sites are omitted from individual analyses.



## 3 Results

### 3.1 Viability of the "predictability" metric

First, we show how the uniqueness and mean performance metrics vary across all models and sites for RMSE, Corr and Overlap. Figure 2 shows each of the 3 metric pairs (rows) for each of the three fluxes (columns), and how those metrics vary

with mean annual temperature in the CRU TS4.01 dataset. Here the uniqueness and mean performance values are similar to those explained in Figure 1, but use more complex models in addition to S_lin (listed in Table 1). Note that uniqueness values less than zero indicate that the local model is not performing better than the global model, as noted above.

– Row 1 shows the RMSE uniqueness of each site, with more unique sites having higher values.

– Row 2 shows the mean of the RMSE of the global and local simulations for each site. For this metric, one might expect
that sites that are more difficult to predict would have higher values, but note that sites with more available energy will generally tend to have larger fluxes and so higher RMSE values, regardless of uniqueness.

– Row 3 shows correlation uniqueness. Like RMSE uniqueness, higher values indicate lower site predictability. Note that there are a large number of zero values for this metric, because for instantaneous linear regression models, correlation is always identical (or inverted) between global and local models, since they are using the same input data, and so
uniqueness is always 0.

– Row 4 shows the mean correlation with observed values for local and global simulations - sites with a low correlation are more difficult to predict (at least by these models). Note that there are a few simulations with 0 mean correlation – these are cases where linear regressions had global and local gradients with opposite sign, resulting in an exactly opposite correlation. In those cases, the zero does not indicate that the global and local simulations had low correlation.

– Row 5 shows the Overlap uniqueness. Higher values indicate sites for which the local Overlap was better than the global Overlap, and negative values indicate the global model performed better in terms of Overlap.

– Row 6 shows the mean model-obs Overlap values of global and local models, and lower values indicate a site that is harder to model in terms of Overlap (Overlap=1 indicates that the model's flux distribution is identical to the observed distribution).

All plots have a fitted generalised additive model (GAM) line, added to help indicate trends in the site means. It is estimated using the pyGAM package (Servén, 2018), using 8 splines, and plotted with a 95% confidence interval.

In Figure 2, we see that there are some patterns in the predictability metrics, which might indicate that mean annual temperature is a driver of predictability, but in general these patterns are not strong. For instance, for RMSE uniqueness (first row), we see a slight increase in uniqueness (or lack of predictability) in sites that are cooler (QUANTIFY?), as well as sites that
have a mean annual temperature around 20°C, for both NEE and Qle. That pattern is less distinct in Qh. There is a stronger trend in RMSE mean (second row) for Qle, but this is likely largely due to the fact that warmer sites naturally tend to have





**Figure 2.** Predictability metrics for mean annual temperature, for all models. The three columns represent the three fluxes, NEE, Qh, and Qle. The six rows show RMSE uniqueness, RMSE mean, correlation uniqueness, correlation mean, Overlap uniqueness, and Overlap mean. Grey points are individual simulation values, blue points are site means across all empirical models. Note that the mean RMSE for NEE is an order of magnitude smaller than for Qh and Qle, and so we have used a different scale for NEE in the second row (Qh and Qle scale indicated on the right).




larger heat fluxes. It seems surprising that Qh does not exhibit the same pattern, since it is more directly related to temperature. Correlation uniqueness (third row) and mean (4th row) shows a similar pattern to RMSE uniqueness for NEE and Qle, where cooler sites and sites around 20°C tend to be harder to predict well. Patterns in Overlap uniqueness (5th row) and mean (6th row) are less clear, but there may be a slight indication of higher uniqueness around 20°C for NEE, and a possible a lower

distribution predictability at higher average temperatures. Note that the negative Overlap uniqueness values are largely due to the fact that regression models do not perform particularly well on extreme values (as indicated in Best et al., 2015).

### 3.2    Determinants of predictability

Since there are a large number of hypotheses to test, only a selection of the most interesting results is shown here. We have also opted to show only the RMSE uniqueness in many plots, since its interpretation is the most straightforward, given the

regression based nature of the empirical models, and since in many cases it correlates with some of the other metrics. Methods and plots for other hypotheses tested are included in the Supplementary Material, along with further details of some of the results presented below (including plots of the other 4 predictability metrics). As some determinants are not available for some sites, the number of site and model combinations in each analysis is noted in each figure title. The figures below use the same methodology as the Figure 2.

### 3.2.1    Predictability as a function of energy and water

The three fluxes we investigate are clearly dependent on the availability of both water and energy. The availability of water is largely defined by precipitation, and temperature provides a proxy for the amount of energy available. We show the RMSE uniqueness for mean precipitation in Figure 3. There appears to be some trend associated with precipitation indicating that the driest sites are more unique for all fluxes, particularly for NEE and Qle.

In Figure 4, the RMSE uniqueness and RMSE mean are plotted as a scatter plot of mean annual temperature and mean annual precipitation. There appears to be some interaction between the two variables, with drier sites with a mean temperature around 20°C showing the highest uniqueness. As in Figure 2, there is also some indication of higher RMSE mean for warmer sites in all fluxes.

Figure 5 shows RMSE uniqueness for aridity index. The pattern shown for each flux, and particularly NEE and Qle, is quite

similar to that for mean precipitation in Figure 3, with more very arid sites being less predictable.

Figure 6 shows how the sites sit in the Budyko framework. The first row shows the sites on a standard Budyko diagram, with actual evaporation divided by mean annual precipitation on the y-axis, and potential evaporation divided mean annual precipitation on the x-axis. Theoretically, a site should fall just below the solid blue line, but location can be affected by available water (e.g. inflow, or precipitation in the period before the measurement period), or the method of estimating potential evapo-

ration. There do not appear to be strong patterns in the potential evapotranspiration uniqueness (see Supplementary material) and actual evapotranspiration appears to have some weak patterns (greater NEE uniqueness at sites with lower evaporation, and the opposite for Qh and Qle, see Supplementary Material), although these are not particularly clear in the Budyko diagrams in the first row. There does not appear to be any pattern in predictability for NEE or Qle as a function of deviance from the





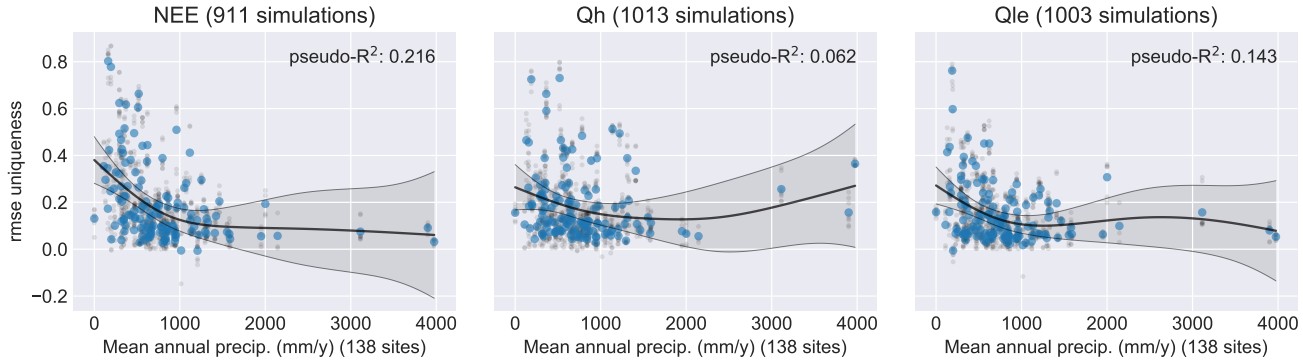

**Figure 3.** RMSE uniqueness for mean annual precipitation.

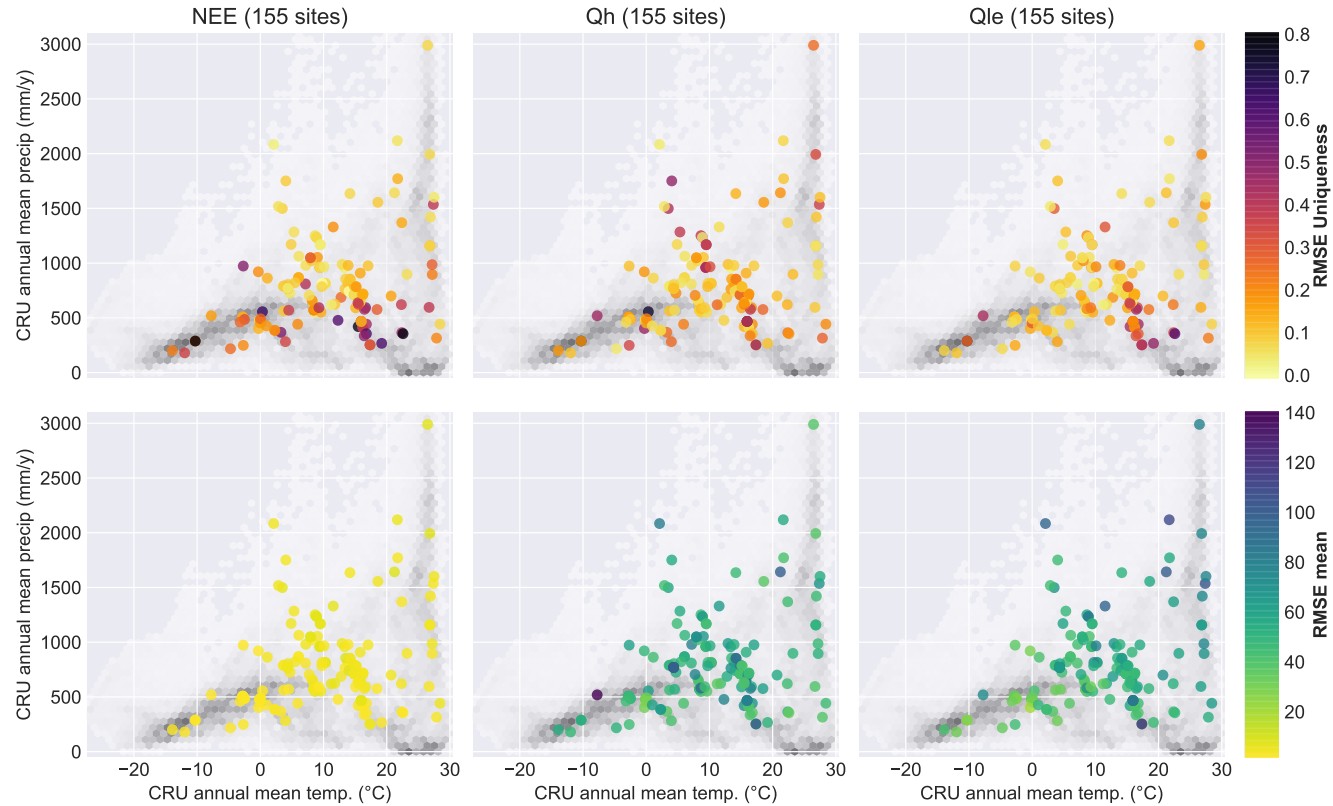

**Figure 4.** Predictability metrics for mean annual temperature vs mean annual precipitation. The top row is RMSE uniqueness (darker colours indicate a more unique, less predictable site), and the bottom row is RMSE mean performance (darker colours indicate higher over-all RMSE). The grey underlying hexbin plot indicates the global distribution of mean precipitation and precipitation from the CRU dataset for all grid cells over land, to give an indication of the representativity of these sites.




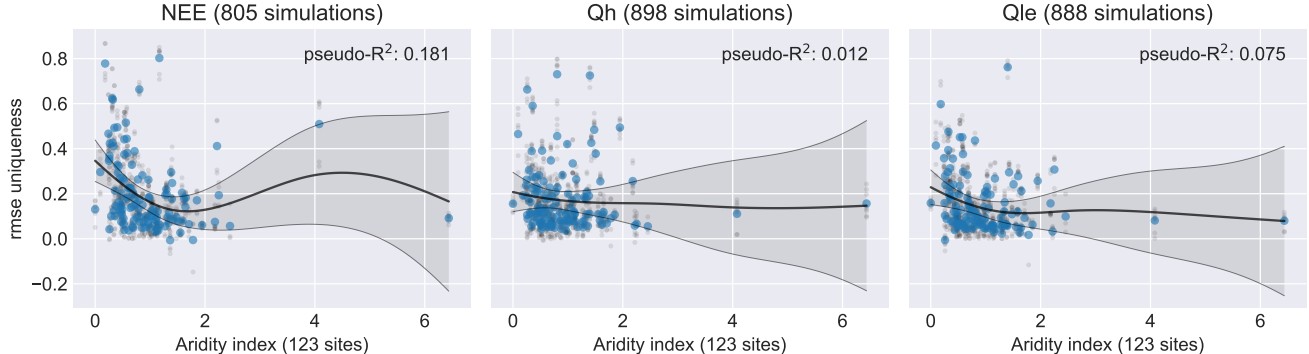

**Figure 5.** RMSE uniqueness for Aridity Index. Two sites with an Aridity Index greater than 3 are excluded for readability, and 32 sites failed the Aridity Index calculation.

Budyko curve (shown in the second row in Figure 6), however there does seem to be some trend toward higher uniqueness for Qh for sites above the Budyko curve (positive deviance). Note that one site (AU-Lox) is excluded from this plot, as its values are too large (AET/MAR of 11.77, and a PET/MAR of 10.72). Its RMSE uniqueness values are 0.352 for NEE, 0.476 for Qh, and 0.438 for Qle. This site and the other sites with AET/MAR values over 2 are all Wetland sites, and as such are likely to
5 have surface water available from upstream run-off in quantities far exceeding that due to precipitation alone.

### 3.2.2 Predictability as a function of site variability

Variability of forcing variables is a major component in the predictability of fluxes. In general, we might expect higher variability to lead to lower predictability. Here we examine predictability at various time scales. Figure 7 shows the RMSE uniqueness over the interannual variability of temperature, and Figure 8 shows the same for precipitation, using the CRU TS4.01 data.
There does not appear to be a strong trend in increased predictability with higher interannual temperature variability Figure 7. However, there does appear to be a clear trend toward higher uniqueness at sites with stronger interannual variability in precipitation for NEE and for Qle (Figure 8).

Other modes of variability descending in scale include intra-annual variability, such as annual range, or variance of monthly values (seasonality); means of particular seasons; and diurnal ranges, as well as mixed-scale measurements, such as isothermal-
15 ity (ratio of diurnal range to annual range of temperature). Measures of each of these for both temperature and precipitation are included in the BioClim data, and plots of uniqueness as a function of each variable are included in the Supplementary Material. We do not include them here because, for the majority of cases, there appears to be no clear patterns of note. The exception includes some increase in RMSE uniqueness in NEE, and perhaps also for Qle, for sites with a higher diurnal temperature range (Figure 9).




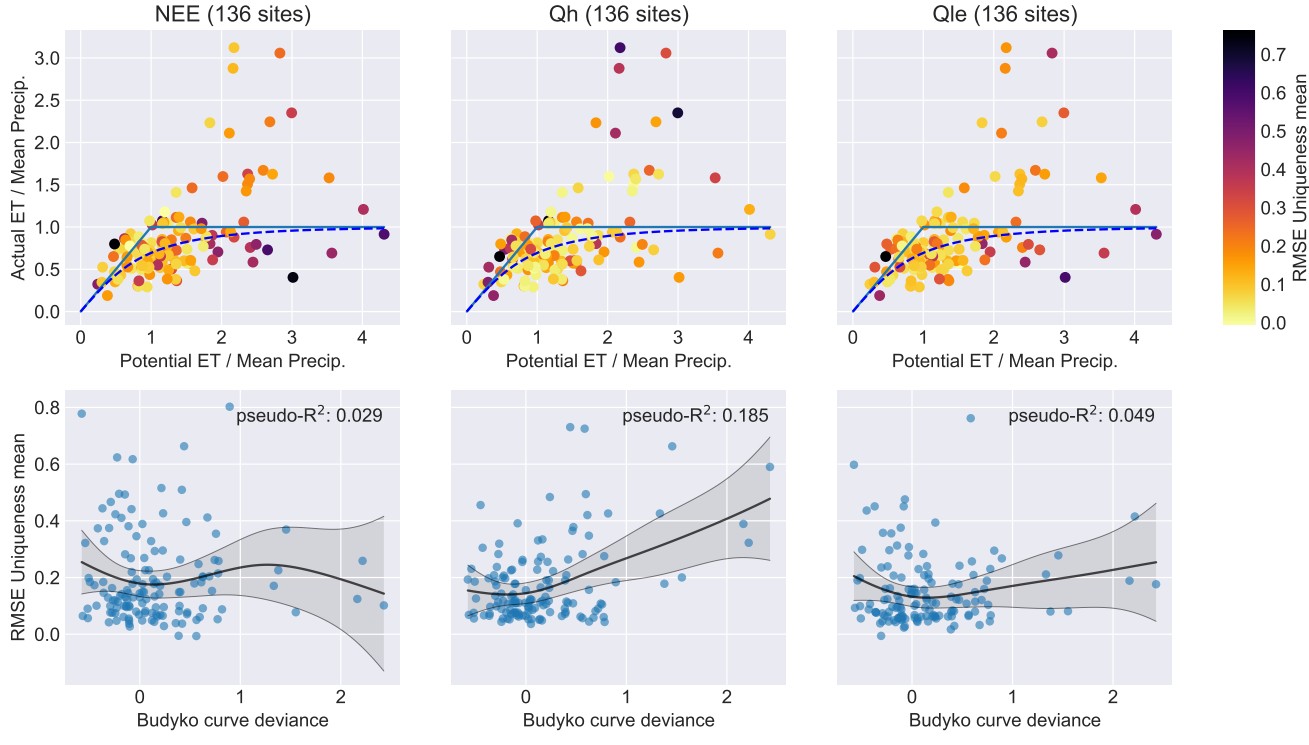

**Figure 6.** RMSE uniqueness for the Budyko analysis. AET and PET come from the FLUXNET 2015 data, mean precipitation comes from CRU TS 4.01. In the first row, colour indicates RMSE uniqueness, averaged across models, where darker colours are more unique. The solid line represents the theoretical energy and water limitations, and the dotted line represents the Budyko curve (Gerrits et al., 2009). The second row shows sites' deviance from the Budyko curve, normalised by the Budyko expectation for the site (sites > 0 lie above the curve in the first row).

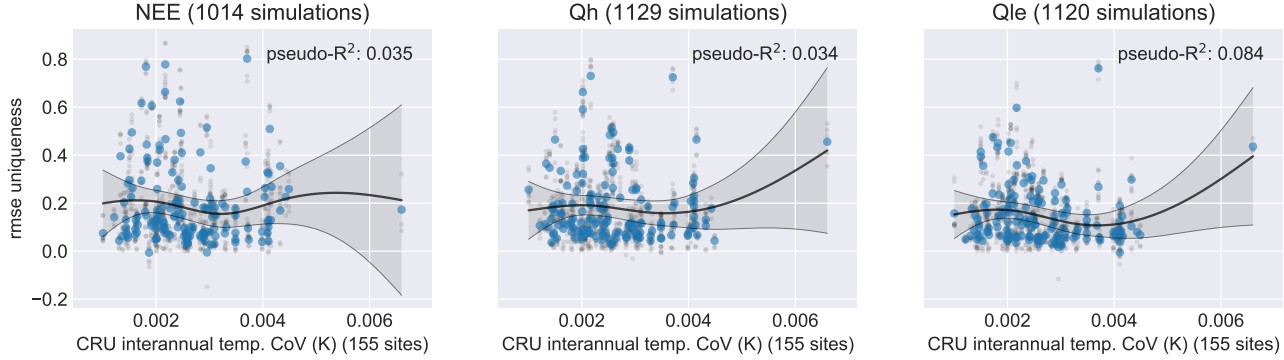

**Figure 7.** RMSE predictability by temperature interannual variability, calculated from the coefficient of variation in the CRU TS4.01 annual means.



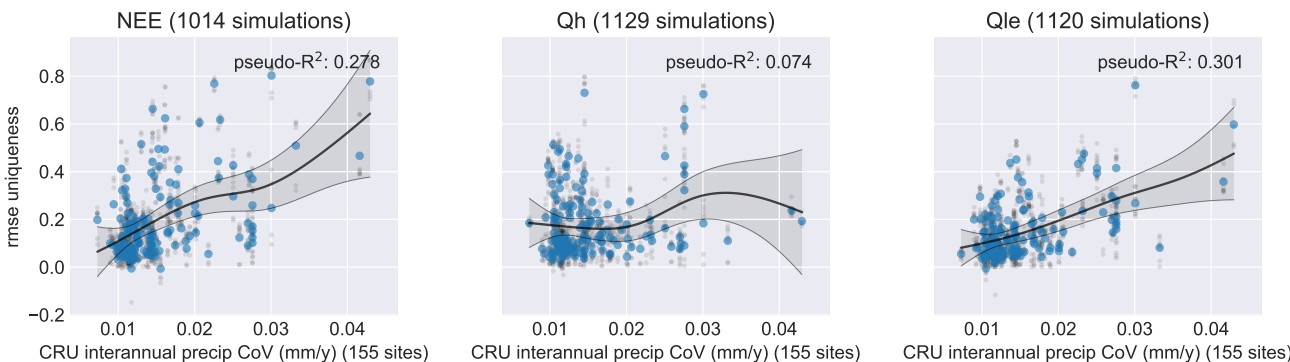

**Figure 8.** RMSE predictability by precipitation interannual variability, calculated from the coefficient of variation in the CRU TS4.01 annual means.

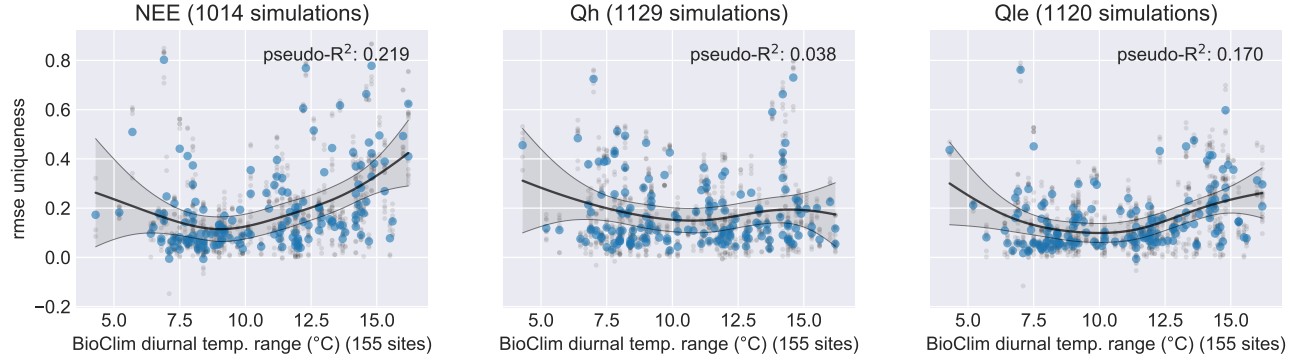

**Figure 9.** RMSE uniqueness for temperature diurnal range.

### 3.2.3 Predictability as a function of vegetation characteristics

Vegetation type is a defining characteristic of different sites, and we would expect different vegetation types to behave differently, reflecting both their adaptations to their environment as well as their response to the met forcing. In particular, we would expect the behaviour of some vegetation types to be more predictable than others. Figure 10 shows the RMSE uniqueness relative to grouped vegetation type (see methods). While there are some differences in uniqueness by vegetation type, few are significant. The main significant differences in RMSE uniqueness (Tukey's honest significant difference test of means across models per site, $p < 0.05$) are:

- For NEE, Shrubland sites tend to be more unique than all other vegetation types.

- For Qh, Wetlands are more unique than Forest types, Shrubland and Savannah, and Grass also tends to be more unique than Evergreen and Deciduous Forests and Savannah.





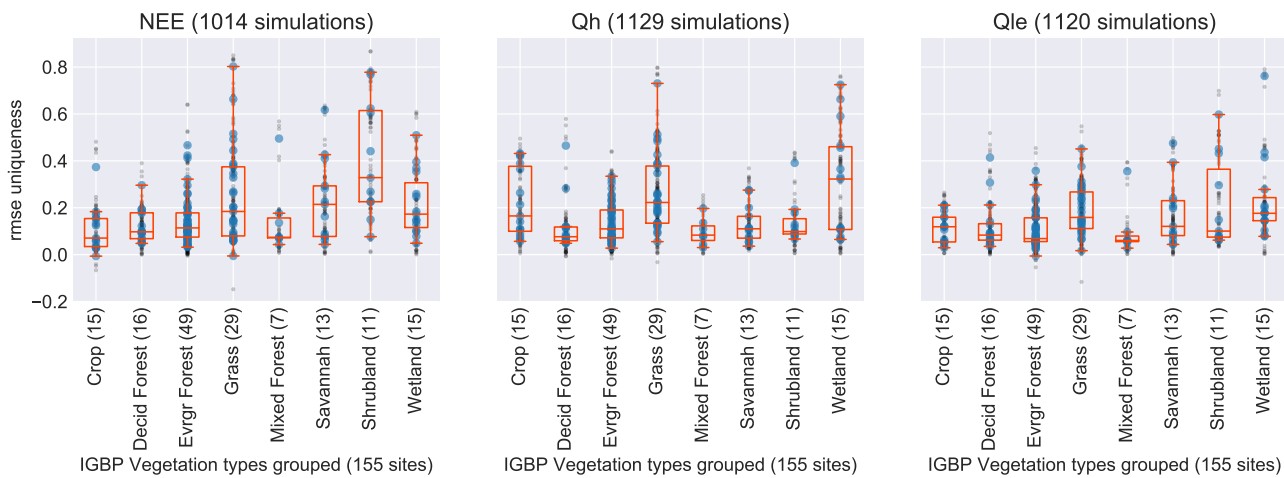

**Figure 10.** RMSE uniqueness for vegetation type (grouped, see Methods).

– For Qle, Wetlands and Grasses tend to be more unique than Evergreen Forests.

However, there is still substantial overlap between even these groups, and the differences between the vegetation type groups are even less distinct when compared over the other five predictability metrics (see Supplementary Material).

### 3.2.4 Predictability as a function of geography

Globally, FLUXNET sites are not evenly distributed, both in space, and in climate regime. Figure 11 shows RMSE uniqueness for NEE as mapped globally, and averaged across models for each site. Given that the models are trained on all sites globally, and those sites are not evenly distributed around the globe (Figure 11) we might expect that sites less well represented (more remote) would be more unique. In Figure 11, there is a hint that more remote sites might be more unique for NEE. Such a pattern is not obvious in the maps for Qh or Qle (see Supplementary Material). To confirm this, we plotted uniqueness by

remoteness (defined as the average distance from a site to all other sites) in Figure 12. There is a indeed a weak trend towards uniqueness at more remote sites for NEE, but not for Qh and Qle. There are no strong patterns evident in remoteness for any variable for any of the other predictability metrics (see Supplementary Material).

### 3.2.5 Predictability as a function of data quality

There are a number of ways that data quality might affect uniqueness. We investigated the energy closure problem in FLUXNET

by comparing predictability as a function of the actual energy closure imbalance, as well as the energy closure imbalance normalised by Rnet. While the energy closure problem in FLUXNET is perhaps one of the most obvious candidates for a determinant of a site's predictability, there does not appear to be a strong pattern in the data for RMSE uniqueness in either plot (nor for any of the other predictability metrics, see the Supplementary Material).





**Figure 11.** Map of NEE predictability - RMSE uniqueness, averaged across models, darker colours are more unique for NEE. In this map, sites are moved to avoid overlap, and a black line joins the site do to its' original location. This way the map gives a better idea of density of FLUXNET in different regions.





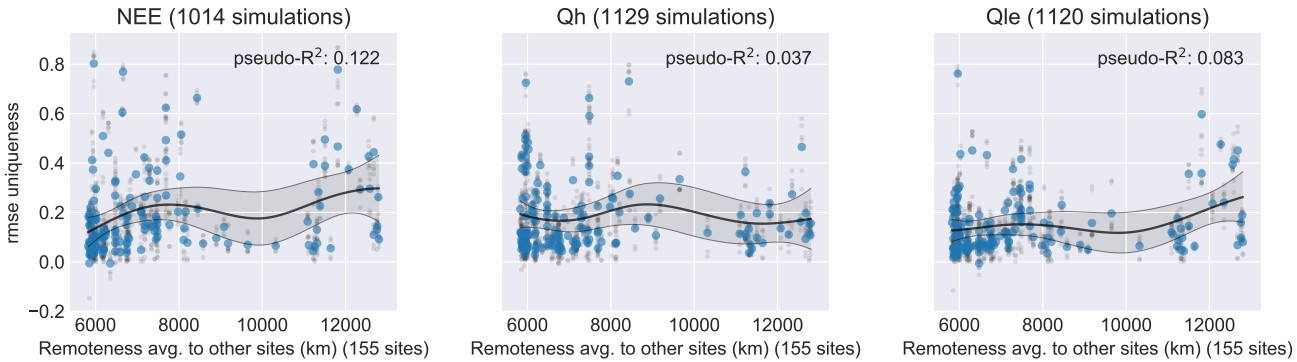

**Figure 12.** RMSE Uniqueness by remoteness (average distance to all other sites)

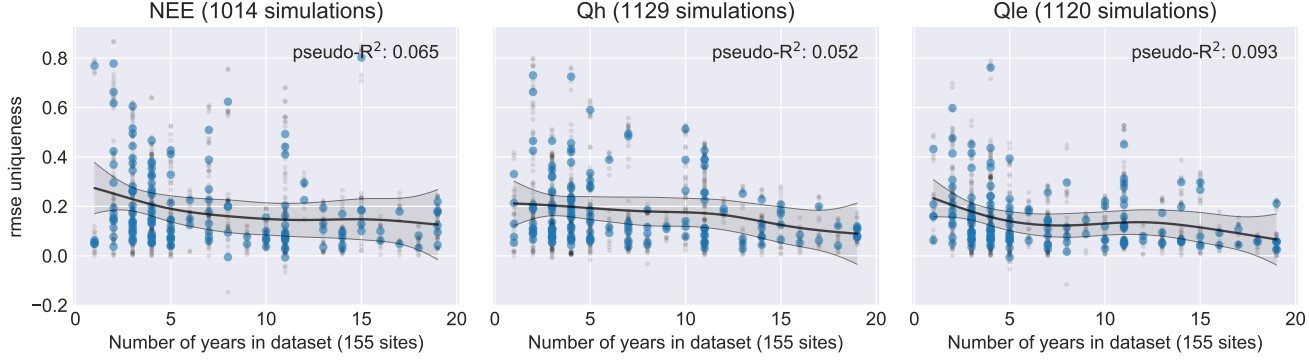

**Figure 13.** RMSE uniqueness for number of years in dataset.

The number of years in the dataset is another obvious candidate determinant of predictability. There does seem to be a weak trend towards shorter sites being more unique, particularly for NEE and Qle (Figure 13). This may be due to longer sites biasing the global training data such that the global model is more like their local models (and hence they appear less unique). This weak trend is somewhat visible in the other predictability metrics (see Supplementary Material, for example in mean Corr, and

5    mean Overlap), but in each case is not strong enough to be significant.

Although the number of years gives a broad scale view of the amount of data in a dataset, it does not tell the whole story. For example, one 2-year site might contain almost a whole 2 years worth of good QC data, while another might contain less than a single year. As such, we also examined the ratio of good QC data to bad QC data at each site. Figure 14 shows the good QC ratio for the flux data combined . Like many of the other potential determinants of predictability, we did not find any clear

10   patterns.




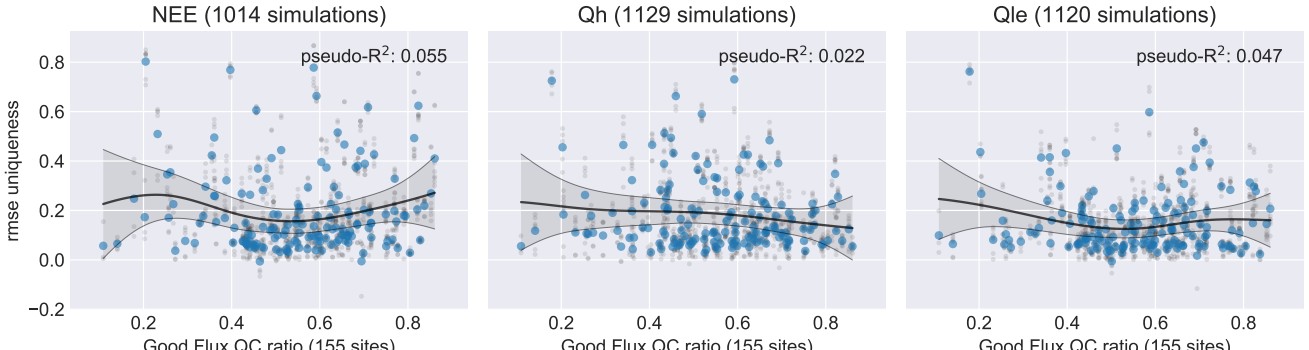

**Figure 14.** RMSE uniqueness for good QC ratio in flux forcings.

### 3.3 Predictability summary

While we have shown that predictability is affected to some degree by various factors (dryness and some vegetation types in particular, it is useful to be able to have an overview of the entire dataset. Figure 15 shows the mean RMSE uniqueness for each of the three fluxes, sorted alphabetically by FLUXNET site code. Here we see that uniqueness is somewhat consistent across

5   variables at each site - Pearson correlation coefficients between variables are: NEE-Qh: 0.113, NEE-Qle: 0.536, Qh-Qle: 0.456. There are interesting differences within clusters of FLUXNET sites, for example the US Metolius sites (US-Me1, US-Me2, US-Me6) are similarly unique for Qh and Qle, but US-Me1 is substantially more unique for NEE, and this site was measured for two years after a fire that killed all trees at the site (Law, 2016). This gives some indication that our uniqueness metric does indeed have bio-physical meaning.

### 10   4   Discussion

In this paper we applied a suite of empirical models to the 155 flux tower sites with half-hourly data included in Tier 1 release of FLUXNET. Our aims were to explore how predictability varied across sites, and then to use this insight into predictability to develop a more systematic approach to guide site selection in model evaluation exercises.

#### 4.1   Site predictability

15   Our multi-site analysis points to marked variability in predictability. For example, it appears that sites in warmer, drier climates tend to be more unique for all fluxes (Figure 3, Figure 4 and Figure 5), and sites with a large diurnal temperature range tend to be more unique, particularly for NEE, and to a lesser extent for Qle (Figure 9). On the other hand, potential determinants that we expected to have quite strong effects on predictability did not appear to do so, for instance mean temperature (Figure 2), dataset length (Figure 13), and major vegetation types (Figure 10). There are several reasons why this might have been the

20   case.





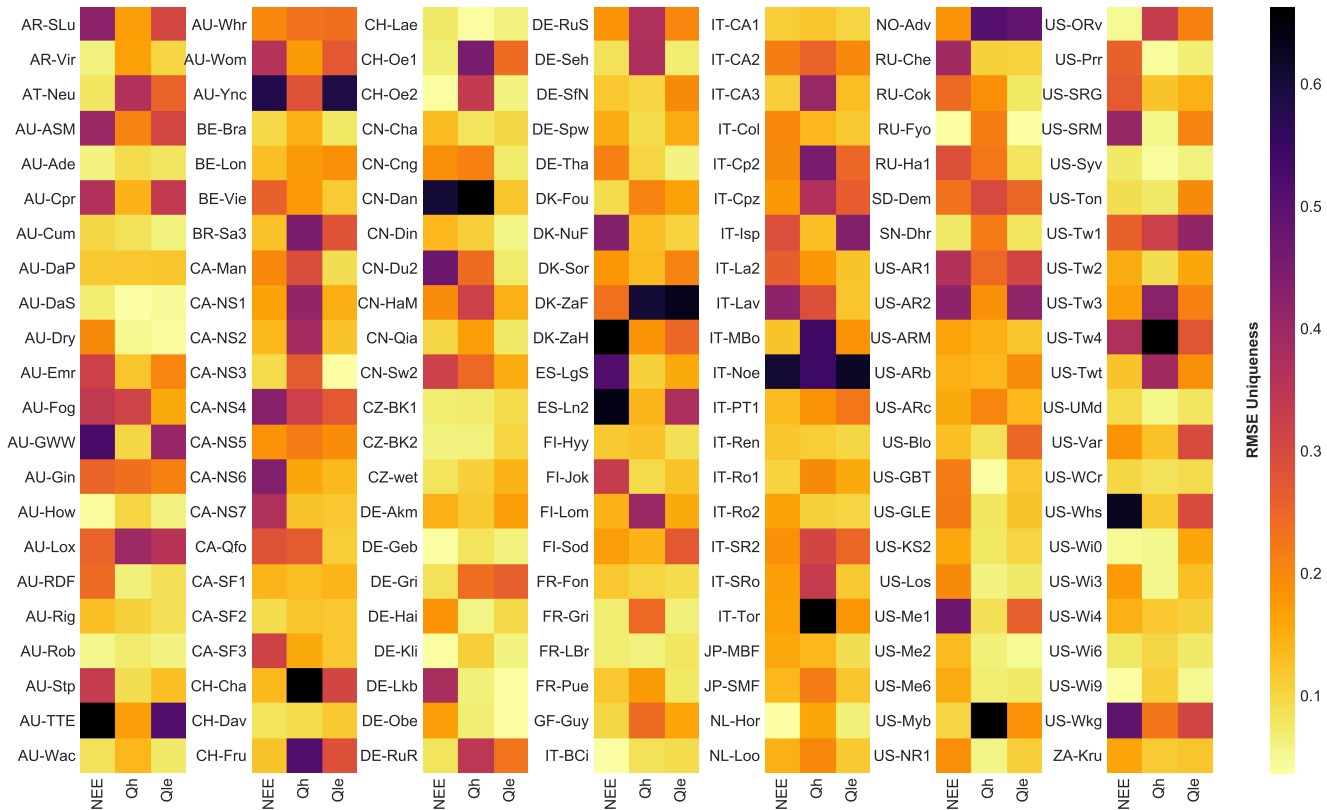

**Figure 15.** RMSE uniqueness mean across models for each flux at each site, in alphabetical order. Darker colours indicate mode unique sites for each flux.

First, the assumption that vegetation type is a major driver of flux behaviour may be wrong. It is perhaps more likely that the widely used approach of analysing FLUXNET sites grouped by a small number of discrete plant functional types is too simplistic, as opposed to exploring differences at a species level, or relating differences to a spectrum of plant traits, plant life spans and metabolism (Kattge et al., 2011; Reich et al., 1997; Wright et al., 2004). Despite widely acknowledged issues with

5    this PFT approach (Alton, 2011; Pavlick et al., 2013; Van Bodegom et al., 2012), this analysis framework is still used, partly because this is the relevant interpretation metric that LSMs use, but also because the necessary information to dig deeper into site differences along these lines is still lacking. Whilst datasets do exist - for example TRY (Kattge et al., 2011), GLOPNET (Wright et al., 2004), LEDA (Kleyer et al., 2008), and ECOFLORA (Fitter and Peat, 1994) - often these are not freely available and the existing ancillary information relating to vegetation available via FLUXNET is minimal, which impedes analyses

10    in this direction. This point was eloquently demonstrated by Konings and Gentine (2016), who used data from the AMSR-E satellite to characterise global variations in isohydricity (the degree to which plants regulate their stomata as leaf water potential



declines). When they categorised their analysis on a PFT level, differences between sites and species were no longer distinct. This remains an avenue ripe for future analysis.

Second, our predictability metrics (RMSE, Corr, Overlap uniqueness and mean) may not be appropriate. There may be systematic biases that inhibit our estimate of predictability due to over-representation of particular biomes, or because mea-
surement periods were not representative. However, the analysis in Figure 12 indicates that there is not a strong trend towards more remote sites being harder to predict, providing some support to our methodology. Our results did indicate a weak trend toward higher uniqueness in sites with shorter measurement periods (see Figure 13), however, a single year of flux data represents a substantial amount of useful data. Short datasets may also be particularly useful if they happen to include rare events that are not well represented in other datasets, such as regional droughts or heatwaves. We nevertheless openly invite construc-
tive arguments against our predictability metric proposal, identification of flaws in the process, or alternative definitions of site predictability or uniqueness.

Should we expect stronger patterns of predictability? In our view, there are strong arguments to support the utility of the FLUXNET data for analyses of predictability. We know that meteorological data measured at flux tower sites does contain a great deal of information about the measured fluxes (Best et al., 2015; Haughton et al., 2018). Indeed the information
contained in the meteorological data about fluxes was very much consistent across sites and this was key to the success of those experiments. So we know that the empirical models used here, which follow a very similar methodology, are capturing the relationships between the meteorological forcing and the predicted fluxes relatively well.

One way we might improve upon this analysis is by focusing on the differences in performance or uniqueness between models with similar structure, but with extra forcing variables. This would tell us something about the predictability contingent
on that variable. For instance, if a model such as STH_km243 (a 243-centre cluster and regression on shortwave down, air temperature, and relative humidity, see Table 1) performs substantially better at a class of sites than an ST_km243 model (the same, but missing relative humidity), then we can say that predictability at those sites may be contingent on information in the humidity data. This analysis is substantially more complex, and so we have left it for future work. The code used to run these models is available at https://github.com/naught101/empirical_lsm, version 1.1 was used for this paper.

**4.2 Model evaluation**

Our second major aim was to develop a more systematic approach for LSM evaluation underpinned by differences in site predictability. Recent work has already illustrated the benefits of defining benchmark levels of performance for a given metric, at a given site (Best et al., 2015; Haughton et al., 2016). The empirical analysis of site predictability we presented goes one step further, effectively quantifying the additional benefit to model performance that site-specific information can provide in
the form of the locally trained empirical models.

Land surface modellers will usually rationalise why a particular module was selected to represent a physical process, or why a specific atmospheric model was used. Given the new information presented in this paper, we suggest that a thorough rationale for why specific FLUXNET sites were used should be explicit in future publications. Importantly, we note that we could not provide evidence that would support site choices based on PFT (Figure 10), data length (Figure 13), quality control (Figure 14)





and so these really do not seem legitimate ways to rationalise choice of sites. We recommend that the predictability of the site is one element for choosing sites, but the process of selecting sites should be more rigorous and reported whether or not this recommendation is followed.

How might this site-specific information be used? Perhaps most obvious would be the clustering of sites, based on their predictability, for use in model evaluation and benchmarking exercises. In Figure 15, we provided some initial guidance to the LSM evaluation community. Here, sites shown in darker colours are sites that exhibit unusual meteorological-flux relationships for a given flux. These are the sites that are likely to present more of a challenge for process-based LSMs to simulate. On the flip side, lighter coloured sites follow commonly observed patterns of behaviour, so good LSM performance at these should be less surprising, and is perhaps less of an achievement. What is important is that modellers should know if the sites they are evaluating their models against are relatively predictable, or unpredictable. Our results, and Figure 15 in particular, gives modellers a tool that can form the basis of a strategy to choose sites, a defence if they choose unpredictable sites and do poorly, and a challenge if they choose more predictable sites and do well. We suspect that the best general strategy for model evaluation would be to pick a set of sites that includes both very predictable sites, as well as very unpredictable sites, with a distribution informed by the determinants of predictability presented above.

Of particular note in Figure 15, but of interest beyond LSM evaluation, is that predictability can be markedly different for different surface fluxes at the same site. For example, we see a number of sites with high NEE and Qle uniqueness, and low Qh uniqueness (e.g. AU-TTE, AU-Ync, ES-Ln2, US-Whs, US-Wkg), and other sites with high Qh uniqueness (e.g. CH-Cha, IT-MBo, IT-Tor, US-Myb). We also see some neighbouring sites with extremely different predictability responses for different fluxes (e.g. DK-ZaF, a wetland site, has very high NEE uniqueness, while the neighbouring DK-ZaH, a heath [grasslands] site, has high Qh and Qle uniqueness). This is evident in other figures where there uniqueness patterns are not shared between fluxes (for example the differences between Qh and the other fluxes in Figures 6 and 9). This provides new justification for different site selection strategies depending on the processes being evaluated.

Our analysis may understandably lead to modelling groups gravitating toward evaluating their models only against a specific sub-sets of FLUXNET sites. We do not think that this is a desirable outcome, and thus have not provided a suggestion of specific sites to use. Indeed care must be taken when evaluating models on small groups of FLUXNET sites due to the greater need to consider the various intricacies of site-specific behaviour. When models are evaluated against a large number of sites, an argument can be advanced that unique site behaviour may average out in the noise. If analysis approaches like ours were to lead to small groups of sites being used to evaluate models, greater care would be needed to capture an adequate diversity of site characteristics. For example, it may be that sites we determine to be unique are simply those that have undergone a disturbance event (e.g. clear felling, fire, wind storms, etc.), or are subject to management (e.g. cropping, irrigation). With improved information about site characteristics (e.g. time since last disturbance), these issues could be avoided. A major advance that would be useful to the LSM community would be the systematic publishing of metadata characterising each site in the FLUXNET data.

Finally, the logical next extension of our work is to evaluate a suite of LSMs at the sites deemed to be most and least pre-dictable, in order to understand the extent to which site predictability translates into model skill. Such an analysis will of course





need careful consideration of the kinds of site eccentricities noted above, noting that information about these eccentricities is not as commonly available as flux and meteorological data. Nevertheless, work of this kind will ultimately help refine how this predictability metric is best utilised in model evaluation strategies.

## 5 Conclusions

In this study, we applied a novel methodology to characterise the predictability of surface fluxes at sites within the FLUXNET2015 dataset. We had two key aims: first, we sought to explain why predictability varied across the 155 FLUXNET sites, with the expectation that we would find patterns in predictability along gradients such as aridity, vegetation type, or in relation to various bioclimatic metrics, both annually and seasonally. Whilst we did show that the 155 FLUXNET sites vary strongly in their predictability, we did not find especially strong patterns in predictability, with the possible exception of aridity. We acknowledge that we might have missed some relevant determinants of predictability, or some transformation of, or interaction between the determinants that we did have available. If we could incorporate these, a clear pattern of predictability might emerge.

Our second aim was to propose a more systematic approach to site selection for model evaluation, underpinned by differences in site predictability. While we found fewer patterns in predictability that we expected, we nevertheless now have a basis on which to define *a priori* expectations of model performance. We suggest that careful choice of FLUXNET sites based on predictability may avoid modellers incorrectly judging their models negatively (via choice of very unpredictable sites) or positively (via choice of very predictable sites). While further work based on this predictability metric is required before a complete rationale for site selection is obvious, we now have a basis on which to develop such a strategy. As a first step, we strongly encourage modelling groups to explain why they choose specific sites for evaluation because, thanks to the FLUXNET community, a lack of availability of data is no longer a reason for site selection.

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

*Competing interests.*

5   *Acknowledgements.* "We acknowledge the support of the Australian Research Council Centre of Excellence for Climate System Science (CE110001028) and the ARC Centre of Excellence for Climate Extremes (CE170100023). This work used eddy covariance data acquired and shared by the FLUXNET community, including these networks: AmeriFlux, AfriFlux, AsiaFlux, CarboAfrica, CarboEuropeIP, CarboItaly, CarboMont, ChinaFlux, Fluxnet-Canada, GreenGrass, ICOS, KoFlux, LBA, NECC, OzFlux-TERN, TCOS-Siberia, and USCCC. The ERA-Interim reanalysis data are provided by ECMWF and processed by LSCE. The FLUXNET eddy covariance data processing

10  and harmonization was carried out by the European Fluxes Database Cluster, AmeriFlux Management Project, and Fluxdata project of FLUXNET, with the support of CDIAC and ICOS Ecosystem Thematic Center, and the OzFlux, ChinaFlux and AsiaFlux offices. A full list of Fluxnet sites and citations is included in the Supplementary material."