# Peer review of "Does predictability of fluxes vary between FLUXNET sites?"

_Biogeosciences, 2018_

## Referee Comment (RC1) · Anonymous Referee #1 · 17 May 2018

The manuscript presents a method for measuring the "uniqueness" of sites based on the ratio of model performance when trained at site level compared to when trained across all sites. I find the rational and motivation for the study to be very relevant, which looks to quantify and empirically examine the the experience that everyone who has worked with diverse datasets such as FLUXNET knows, that it is easier to get good model performance as some sites than others. However, I find the manuscript in it's current state to be rather unfocused, needing more synthesis to focus on key hypothesis and findings and focusing on what the metrics can explain (not what they can't explain) and what is most useful for the users of FLUXNET and other datasets. There are some good outlines of possible avenues for analysis on page 24, lines 18-24, which are discounted as being too complex. While I can appreciate that this synthesis

work is complex, the lack of a clear message really hinders the usefulness of the paper as is.

Some possible ideas:

- Sensitivity of the metric such as within the empirical ensemble: how dependent is the metric on model used, QC of predictor variables in the empirical models, site selection when calculating the metric: does it change drastically if the global run is only performed on a subset of sites (how many sites are needed?).

- Based on your analysis, what are the most and least unique sites with respect to each flux?

- Possibly framing the analysis as a variable selection framework to predict uniqueness.

Abstract

Nearly half of the abstract is motivation, while containing no tangible results or discussion.

- Pg 1, line 12: "A number of hypotheses potentially explaining site predictability were then tested..." This is very vague, could at least give number of hypothesis, or focus on those that are most important.

Introduction

- Pg 2, line 3: "Perhaps surprisingly, the predictability of a site is rarely considered when choosing sites to evaluate models." I would argue this is the key motivation, but also with the caveat that predictability is likely considered (either explicitly or implicitly) but not quantified and often not discussed.

- Pg 2, lines 5-12: While I can appreciate that the study is motivated from a LSM perspective, they are not evaluated in the manuscript, making this paragraph unnecessary. Furthermore, predictability may be useful in other contexts such as empirical upscaling of fluxes (Tramontana et al 2016).

- Pg 2, lines 35-36: "...were not able to identify any obvious patterns in model performance across sites." patterns with respect to what?

- Pg 3, lines 8-24: Is there an indication that any of these studies would potentially have a more/less predictable subset of sites? Not to pick on any one study, but are there any indications they could have a selection bias that would benefit from the uniqueness metrics you are proposing?

- Pg 4, line 3: Does the method presented here not have the same assumption? For example, if the empirical model was a random number generator the RMSE between local and global would be the same and uniqueness would always be 0?

- Pg 4, lines 7-11: Here the outline tells that the manuscript will both be an exploration analysis ("investigate several hypothesis"), but then promises a sound theoretical basis for site selection. It would be useful to outline how the hypothesis you explore will lead to concrete methods that are useful to modelers, because at current state the manuscript requires a fair amount of digging in order to get any idea as to what sites will be more or less predictable.

Methods

- Pg 4, line 24: The empirical models really need to be explicitly describe in the manuscript, seeing as they are the basis for calculating your metric. Furthermore, is it necessary to use this particular suite of models or will any empirical model do?

- Pg 4, line 29: This is also a subset as you do not have infinitely many sites. How robust is the metric to site selection. How variable is the metric when performed on one subset of sites to the next?

- Pg 5, line 10-25: It seems this work is very reliant on the previous works (Best et al. (2015), Haughton et al. (2016), and Haughton et al. (2018)) and as such these studies should be outlined more. In the current state, one would need to read the previous three papers to understand the core methodology presented in this manuscript. For

instance, there is no citation for the cluster-plus-regression methodology. Furthermore, the long$_{term}$ and short$_{term}$ notations are never introduced.

- Pg 6, Table 1: Did the long$_{term}$ 729 model never have a successful simulation? In which case it was never actually used? Also, f

- Pg 6, line 1: How would one interpret the "Mean performance" metric? Is there an advantage of this method compared to simply taking the arithmetic mean of local and global RMSE rather than the distance from the origin?

- Pg 7, line 10: Does this suggest that the uniqueness metric, when using the different QC flags between the training and prediction runs, is combining both the effects of information content of the predictor variables and the gap filling? As you state the difference can be large with the complex models, as such this should be reported. Would it make sense to always use the same QC flags for your analysis?

- Pg 7, line 17: Possibly using a combined summary statistic could simplify the procedure a bit? e.g. Gupta et al 2009

- Pg 7, line 24: Was there any attempt to prevent over-fitting, such a cross validation. How resistant is the cluster-plus-regression model to over-fitting?

- Pg 8, lines 1-10: Again, a sensitivity analysis of how model and QC selection effect the metric would give users more confidence in the metrics.

- Pg 8, line 8: While these caveats have been raise, really some effort to test the impacts of each should be included in the manuscript. I think it would be more beneficial to show the sensitivity of the metric rather than the extensive hypothesis testing.

- Pg 9-11: While I appreciate the thorough analysis, I feel that the paper is lacking focus and comes across as sort of a data-dump. Especially given the fact that many of the results are inconclusive. Possibly focusing on some of the most promising hypothesis and moving many others to the supplemental material, or removing and simply mention that they were tested and the results were inconclusive. In all honesty I had to take a break from reading the paper after finishing this section.

- Pg 12, line 5: Is there a reason for using the CRU mean annual temperature and not the mean annual temperature from the sites themselves?

- Pg 12, line 29: Seems an editing note got left in "(QUANTIFY?)". It would probably be a good idea to quantify what you mean by cooler.

- Pg 13, Figure 2: There seems to be some anti-correlation between the uniqueness and mean metrics, particularly with the RMSE of NEE. Is this likely just spurious?

- Pg 14, line 4: Typo: "a possible a lower", furthermore, it would be beneficial to be more exact, as the results often report that there may be patters.

- Pg 15, Figure 4: The use of two colormaps with overlapping colors can be confusing, giving a false indication that the yellow in both plots is related.

- Pg 16, line 9: Again, using CRU for precipitation data when you have site level data seems curious.

- Pg 16, line 18: Is high diurnal temperature range not related to mean temperature? Can you differentiate this signal from that seen in Figure 2?

- Pg 17, Figure 6: Deviance from Budyko curve is never explicitly defined.

- Pg 18, line 2: "met forcing", met. is an abbreviation.

- Pg 18, line 9: "Shrubland and Savannah, and Grass", => Shrubland, Savannah, and Grass?

- Pg 22, line 2: parentheses has no close.

- Pg 22, line 8: This is an example of a concrete example which give more confidence in the metric, yet it is given little attention compared to other analysis which are relatively inconclusive. If other examples exist possibly they could be highlighted.

- Pg 23, Figure 15: Could this figure be organized in a way that gives more information, such as ordering by uniqueness or grouping by PFT? In the current state it would

maybe be more useful as a table with actual numbers.

- Pg, 24, line 5-7: I don't follow your logic here. I am not sure how the lack of a strong trend in Figure 12 provides support to the methodology. Also, I would not conflate the proximity of one tower to other towers with biome representativeness.

Gupta, H.V., Kling, H., Yilmaz, K.K., Martinez, G.F., 2009. Decomposition of the mean squared error and NSE performance criteria: Implications for improving hydrological modelling. Journal of Hydrology 377, 80–91. https://doi.org/10.1016/j.jhydrol.2009.08.003

Tramontana, G., Jung, M., Schwalm, C.R., Ichii, K., Camps-Valls, G., Ráduly, B., Reichstein, M., Arain, M.A., Cescatti, A., Kiely, G., Merbold, L., Serrano-Ortiz, P., Sickert, S., Wolf, S., Papale, D., 2016. Predicting carbon dioxide and energy fluxes across global FLUXNET sites with regression algorithms. Biogeosciences 13, 4291–4313. https://doi.org/10.5194/bg-13-4291-2016

---

## Referee Comment (RC2) · Anonymous Referee #2 · 18 May 2018

This manuscript presents a methodology for quantifying the "predictability" of land atmosphere fluxes of water, energy and carbon across 155 eddy covariance sites, with the goal of helping to better interpret comparisons between these observations and output from land surface models. This idea has considerable merit, and could be of interest to a large number of land surface model developers, and other synthesizers of eddy covariance data sets. Unfortunately, in its current form it is difficult to extract the most important information, as there is insufficient emphasis on what might be valuable, and much material is included which is not relevant. Overall, it needs to be much more focused, and the authors need to concentrate on: (i) predictability and their predictability metric; (ii) the models used, whilst greatly streamlining the hypotheses, given the inconclusiveness of the majority of the analysis.
General Comments

1. As the authors acknowledge, "there is no single definition of predictability" – and this is a key challenge to this paper. The introduction needs to address this much earlier than the bottom of page 3, so there aren't several pages of text discussing something that has not been defined, or at least how it is being treated in the context of this manuscript.

2. It seems that the authors are treating predictability as the inverse of "uniqueness", which is characterized as the deviation between a globally optimized model, versus a locally optimized model. This needs additional clarification and justification. With this definition, the predictability is inherently model dependent, rather than some intrinsic property of the site alone. This is always the case possibly, but needs to be spelt out. It also highlights the importance of the models.

3. A reliance on this uniqueness as a proxy for predictability seems like it might have drawbacks. Consider the NEE plots in Figure 1. There are a group of sites with relatively high "uniqueness", and thus low predictability, but with a global RMSE less than 7, which is lower than for a large fraction of sites indicating a better mean performance. Are these sites more or less predictable? It seems that this can only be quantified through combing the uniqueness metric with the mean performance into a single metric, but uniqueness is discussed in isolation throughout the results section.

4. This can be addressed by combining the metrics with appropriate weights. The authors say this is not done due to the difficulty in combining the different metrics, but given the lack of information in the metrics other than RMSE, and the apparent requirement to combine uniqueness and mean performance, this should be reconsidered, with at least these two components of each metric combined.

5. What is the additional information gained from switching from Cartesian to polar coordinates? Would not a simple mean of the global and local models, and the normalized difference between then suffice?

6. This manuscript relies heavily on previous work (Best et al, 2015, Haughton et al., 2018). Indeed, it is not possible to understand much about the models with out consulting this closely. Given how dependent the predictability metrics are on these models, some further description of them is required here.

7. Indeed, it's very unclear why multiple models are being used at all? What is the benefit of doing this rather than using the single "best" model?

8. Although the authors suggest they want to leave this for future work as it is "substantially more complex", it seems at least some examples are required to explain how the predictability metrics are sensitive to the models, and how this can be interpreted when discussing specific sites. For example, it seems that many semi-arid sites are characterized by these models to have high uniqueness, but what would happen if soil moisture was included in the model?

9. Whilst uniqueness as defined here is certainly a useful metric to assess flux sites by, and to help interpret comparisons between observations and land surface model output, it is unclear it represents something like the inverse of "predictability". In fact, a contrary argument could be made that the sites that exhibit large reductions in model error when optimized with local data are the most predictable. Whilst for those sites that don't see model improvement when just local data are used this lack of sensitivity might also be interpreted as a lack of predictability, particularly for sites with low mean performance. In this context, a predictable site is one where given more information, model skill increases, and whilst at an unpredictable site specific information does not increase skill.

10. Both sections 2.2 and 3.2 read as an overly long laundry list of "everything we tried". It will be easy to greatly increase the overall focus of the manuscript by addressing this. Given the lack of conclusiveness regarding the majority of the hypotheses about determinants of predictability, a brief note that they were considered and findings were inconclusive is all that is required.

Specific Comments

P1 L1-8. Rather like the manuscript as a whole, the abstract needs much more focus. Emphasis specific detail, not background information and motivation.

P4 L13-21. Not methods.

P8 Fig 1. Don't understand the need for colored dots?

P12 L29. Yes, you might want to [QUANTIFY] that

P15 Fig 4. Presumably it is the mean values that are being plotted here?

P15 Fig 4. Seems like NEE needs a different scale?

P16 Fig 5. The two sites with an aridity index higher than 3 haven't been excluded.

---

## Author Comment (AC1) · 20 Jun 2018

> The manuscript presents a method for measuring the "uniqueness" of sites based on the ratio of model performance when trained at site level compared to when trained across all sites. I find the rational and motivation for the study to be very relevant, which looks to quantify and empirically examine the the experience that everyone who has worked with diverse datasets such as FLUXNET knows, that it is easier to get good model performance as some sites than others. However, I find the manuscript in it's current state to be rather unfocused, needing more synthesis to focus on key hypothesis and findings and focusing on what the metrics can explain (not what they can't explain) and what is most useful for the users of FLUXNET and other datasets. There are some good outlines of possible avenues for analysis on page 24, lines 18-24, which are discounted as being too complex. While I can appreciate that this synthesis work is complex, the lack of a clear message really hinders the usefulness of the paper as is.

We are glad the reviewer found the concepts in the paper relevant, and thank them for their comprehensive comments. We agree that the paper needs a tighter focus, and have found the reviewers comments valuable in making those improvements. The reviewer also presents many useful ideas for extended analysis to make the method more rigorous, and we have attempted to implement as many ideas as we could, without expanding the paper dramatically.

We note in particular the Reviewer's concerns about site and model selection, and the possibility of variance in the results. We have added some analysis to more adequately assess the impact of the multiple models, and made a number of other changes in an attempt to add some clarity to the paper.

Responses to individual comments are provided below.

**Some possible ideas:**

> Sensitivity of the metric such as within the empirical ensemble: how dependent is the metric on model used, QC of predictor variables in the empirical models, site selection when calculating the metric: does it change drastically if the global run is only performed on a subset of sites (how many sites are needed?).

We have extended the analyses of each hypothesis by adding two separate GAM plot lines to each GAM figure, one for each of two subsets of models: the two simple linear models, and the models with longer-term averages included (short_term243, long_term243). These serve to show that while there are certainly differences between the models in terms of performance - the more complex models do perform substantially better in the mean, and also pick up more uniqueness - the relative differences between the sites for the different model types remains qualitatively the same in all cases. We have added the following to the paragraph describing the GAM plots:

*We have also fitted two other GAM models using subsets of the model ensemble: In each such plot, the red line represents a GAM fit using only the linear regression models (S_lin, ST_lin), and the purple line represents only the models with lagged input variables ("Longer models" - short_term243, and long_term243). These serve to show any differences in the predictability metrics that are contingent on model complexity, non-linearity, or input variables.*

We have also added text to figure captions and results text where relevant, in particular to the first GAM plot. We added the following to the paragraph describing this plot:

*We note that the two subset GAM plots, here and in later plots, describe a similar pattern in each metric in most panels. The main differences seem to be largely to do with the more complex models' ability to capture more of the variance: the mean performance of these models under each metric is better (and the linear models' worse) than the mean, and the uniqueness is higher for the Corr and Overlap metrics, but quite similar for RMSE.*

We don't think it would be useful to run separate analyses for different sets of QC flags, as discussed below in response to the comment on Pg 7, line 10.

The Global run should not change drastically based on a random subset of sites. The nature of the data is similar at each site globally, and we are using regressions with few parameters, relative to the number of data points, so there is very little chance of over-fitting the global dataset.

> Based on your analysis, what are the most and least unique sites with respect to each flux?

This question is answered in Figure 11, and we have modified it further with numerical values to make the differences between sites explicit. In writing this paper, we considered making site recommendations, but ultimately decided that such statements could end up being more misleading than useful. Predictability is potentially a very useful metric to have available when deciding on sites to use for a study, but ultimately each study will have different aims, and other factors such as vegetation representation and climatic zone will also play a large decision in site selection, which will be tailored for those aims. We have added a sentence to this effect in the second paragraph of the last discussion section:

*We intentionally avoid recommending a particular set of most or least predictable sites, as the suitability of a given set of sites for a particular study is going to be dependent on many factors.*

> Possibly framing the analysis as a variable selection framework to predict uniqueness.

This study is not a variable selection framework. Haughton et al. (2018) provided the variable selection framework that this study uses.

**Abstract**

> Nearly half of the abstract is motivation, while containing no tangible results or discussion.

> Pg 1, line 12: "A number of hypotheses potentially explaining site predictability were then tested…" This is very vague, could at least give number of hypothesis, or focus on those that are most important.

We agree, and have reduced the amount of motivation, and included some more specifics about results in the abstract, which now reads:

*The FLUXNET dataset contains eddy covariance measurements from across the globe, and represents an invaluable estimate of the fluxes of energy, water and carbon between the land surface and the atmosphere. While there is an expectation that the broad range of site characteristics in FLUXNET*

*result in a diversity of flux behaviour, there has been little exploration of how predictable site behaviour is across the network. Here, 155 datasets with 30 minute temporal resolution from the Tier 1 of FLUXNET2015 were analysed in a first attempt to assess individual site predictability. We defined site **uniqueness** as the disparity in performance between multiple empirical models trained globally and locally for each site, and used this along with the mean performance as measures of predictability. We then tested how strongly uniqueness was determined by various site characteristics, including climatology, vegetation type, and data quality. The strongest determinant of predictability appeared to be that drier sites tended to be more unique. We found very few other clear predictors of uniqueness across different sites, and in particular little evidence that flux behaviour was well discretised by vegetation type. Data length and quality also appeared to have little impact on uniqueness. While this result might relate to our definition of uniqueness, we argue that our approach provides a useful basis for site selection in LSM evaluation, and invite critique and development of the methodology.*

**Introduction**

> Pg 2, line 3: "Perhaps surprisingly, the predictability of a site is rarely considered when choosing sites to evaluate models." I would argue this is the key motivation, but also with the caveat that predictability is likely considered (either explicitly or implicitly) but not quantified and often not discussed.

We agree that this was poorly worded, and have changed the sentence to read:

*Perhaps surprisingly, the predictability of a site is rarely considered explicitly when choosing sites to evaluate models.*

> Pg 2, lines 5-12: While I can appreciate that the study is motivated from a LSM perspective, they are not evaluated in the manuscript, making this paragraph unnecessary. Furthermore, predictability may be useful in other contexts such as empirical upscaling of fluxes (Tramontana et al 2016).

This is a good point. We have removed the irrelevant historical assessment, and focussed on the more important point that PFTs are an unknown factor, as far as predictability goes. The paragraph now reads:

*Modern land surface models (LSMs) attempt to describe the exchange of energy, water and, more recently, carbon, by explicitly representing the soil-vegetation continuum (Pitman, 2003). Common to virtually all LSMs is an assumption that flux behaviour variations between biomes, given similar driving conditions, can be explained by a small sample of structural and physiological parameters, grouped as plant functional types (PFTs). As a result, land modellers have sought observations from locations characteristic of these broad PFTs to develop and evaluate models. However, the actual practical representativeness of PFTs of the underlying vegetation properties has only recently begun to be investigated (e.g. Alton, 2011), and no explicit empirical assessment of PFTs as a driver of predictability has been undertaken.*

> Pg 2, lines 35-36: "...were not able to identify any obvious patterns in model performance across sites." patterns with respect to what?

With respect to differences between sites. However, we have decided to remove this sentence. It was a distraction from the main message of the paragraph.

> Pg 3, lines 8-24: Is there an indication that any of these studies would potentially have a more/less predictable subset of sites? Not to pick on any one study, but are there any indications they could have a selection bias that would benefit from the uniqueness metrics you are proposing?

We do not know, and we would prefer not to speculate, as this is not really knowable without replicating those studies on other datasets. However, this is an interesting question, and we have added the following sentence to that paragraph to address it:

*Whether or not any of the studies mentioned above are biased by a lack of consideration for predictability is unknown, because this this was not part of the selection process for the sites chosen.*

> Pg 4, line 3: Does the method presented here not have the same assumption? For example, if the empirical model was a random number generator the RMSE between local and global would be the same and uniqueness would always be 0?

Yes, it would, on average. We agree the that the metric is only useful when used with models that have some predictive power. The Kaboudan (2000) metric is also univariate, so we have changed that sentence to read:

*Kaboudan (2000) provides another univariate predictability metric.*

> Pg 4, lines 7-11: Here the outline tells that the manuscript will both be an exploration analysis ("investigate several hypothesis"), but then promises a sound theoretical basis for site selection. It would be useful to outline how the hypothesis you explore will lead to concrete methods that are useful to modelers, because at current state the manuscript requires a fair amount of digging in order to get any idea as to what sites will be more or less predictable.

We disagree that this section promises anything - it simply sets out aims. However, in an attempt to improve clarity, we have changed the last sentence of the introduction to read:

*This will allow expectations of model performance to be better defined by providing a priori estimates of local predictability based on site characteristics. We hope this can provide some mitigation of the potential for ad-hoc site selection to shape judgement of how well LSMs perform.*

**Methods**

> Pg 4, line 24: The empirical models really need to be explicitly describe in the manuscript, seeing as they are the basis for calculating your metric. Furthermore, is it necessary to use this particular suite of models or will any empirical model do?

We removed the parenthetical remark about the models from this paragraph, so that the paragraph is model-agnostic.

See also our response to Pg 5, line 10-25 comment.

> Pg 4, line 29: This is also a subset as you do not have infinitely many sites. How robust is the metric to site selection. How variable is the metric when performed on one subset of sites to the next?

As noted above in response to the first comment, we expect the global simulation to be very stable, due to the nature of the models, and the size of the dataset. The metric itself should be more stable due to being an average across multiple models. The metric will vary between sites as the local performance varies, but this is the point of the exercise.

> Pg 5, line 10-25: It seems this work is very reliant on the previous works (Best et al. (2015), Haughton et al. (2016), and Haughton et al. (2018)) and as such these studies should be outlined more. In the current state, one would need to read the previous three papers to understand the core methodology presented in this manuscript. For instance, there is no citation for the cluster-plus-regression methodology. Furthermore, the long_term and short_term notations are never introduced.

The beginning of the 4th paragraph of the Methods already describes the cluster-plus-regression models - they are conceptually simple, and there is not a lot more to say, but we have split this paragraph in two, and changed the first part to add some description of the use of the cluster-plus-regression models:

*This procedure is model-agnostic, and we have used models in the framework developed in Best et al. (2015) and Haughton et al. (2018), because they are conceptually simple, but able to fit complex functional relationships. These models (listed in Table 1) include some simple linear regressions, as well as cluster-plus-regression models. The cluster-plus-regression models consist of a K-means clustering over meteorological driving data, and then an independent linear regression between drivers and fluxes at each cluster. These cluster-plus-regression models can fit arbitrary functional forms between predictor and response variables, when using a high enough cluster count (k), and given enough data. The models are not perfectly deterministic, since K-means convergence is dependent on cluster initialisations, but the variance in the results is small (see supplementary material, Haughton et al., 2018), and unlikely to substantially affect our results substantially. Our use of an ensemble of models at each site further mitigates this problem. The ensemble also allows us to overcoming the problems of the simpler models failing to capture behavioural nuances, and of the more complex models failing to train at some sites due to insufficient data (described below).*

We also moved the note about which fluxes are modelled to the top of the methods section.

We have added the long-form of each long-term/short-term model name to the table.

> Pg 6, Table 1: Did the long_term_729 model never have a successful simulation? In which case it was never actually used? Also, for the simple models it seems that negative uniqueness happens quite frequently for corr and overlap, [unreadable in PDF]

This is correct. To save on confusion, we have removed all mention of the long_term729 model from the paper, and adjusted text to suit.

The second part of this comment was truncated in the review PDF, so we may have missed part of it. Yes, these metrics suffer more from negative uniqueness with the simple models.

We have looked more into the cause of the overlap in correlation, and it appears that most of the problem in correlation is very small - all of the negative correlation uniqueness are > -10e-14 for S_lin. We have set a negative uniqueness threshold of -1e-8 instead of 0 in all cases, and updated the table to reflect that, and added a note to the table caption.

Some are down to -1.7e-2 for ST_lin which can be accounted for by the different weights on the two input variables.

For the Overlap metric it is possible for the global model to produce a stronger trend, e.g. because the training dataset includes more diversity relative to noise. This would result in a higher variance, which would counteract the smoothing effect of the linear regressions, and produce a better overlap score for the global model, resulting in negative uniqueness. We added the following sentences to the paragraph that mentions the negative uniqueness section of the table:

*We note that ST_lin Corr has a relatively large number of negative uniquenesses, which can be accounted for by better estimates of the S and T variable coefficients in the global model. There are also negative Overlap uniquenesses in the linear models, which is likely due to the global model training resulting in a stronger trend, and thus a higher variance, counteracting the fact that empirical models are generally smoothers.*

> Pg 6, line 1: How would one interpret the "Mean performance" metric? Is there an advantage of this method compared to simply taking the arithmetic mean of local and global RMSE rather than the distance from the origin?

It is the same thing. We have changed the sentence to read:

*Mean performance: the arithmetic mean of the local and global metric at each site, defined by distance from the origin.*

> Pg 7, line 10: Does this suggest that the uniqueness metric, when using the different QC flags between the training and prediction runs, is combining both the effects of information content of the predictor variables and the gap filling? As you state the difference can be large with the complex models, as such this should be reported. Would it make sense to always use the same QC flags for your analysis?

This is a problem that we struggled with. Yes, it is combining the effects of predictor variables and gap-filling. We added this sentence to the end of the paragraph:

*We considered the option of using the training QC flags for the evaluation period, however this would result in different models having very different evaluation periods.*

We have also updated Figure 1, and adjusted the caption to reflect this. Now, row 1 shows all of the data that was in both rows, using "tail" lines to join the dots. We have added another row using short_term243 as an example of a more complex model. This figure now more clearly shows the variance added to the RMSE due to the discrepancy between the flux-only and met+flux QC flags. We have updated the paragraph in text to describe this, and the implications shown in the graph:

*The uniqueness and mean performance metrics are shown for RMSE in Figure 1 for the S_lin and short_term243 models to illustrate how to interpret later figures:* uniqueness *is the angle measured clockwise from the origin (the optimal metric value) and the 1:1 line (equal local and global performance), and* mean performance *is the average performance of the local and global simulations, given by the distance of each point from the origin. Each point is a different site. Figure 1 also illustrates the differences between the results when the local training data is identical to the testing data, and when it differs due to mismatch between the meteorological and flux QC flags between training and testing. In each panel, the blue points indicate the local and global RMSE values used for the simulation in the remainder of the study. The tail from each point indicates where these values would have been if the same QC data that was used for training was used for evaluation (meteorological + flux QC, instead of just Flux QC. The tail points are strictly at or below the*

*1:1 line (as the empirical fit is optimised for RMSE locally, but not globally). The flux-only QC evaluated blue points can shift, and some lie very slightly above the 1:1 line. Tails pointing towards the origin indicate that these simulations' mean RMSE is worse than it would be using the training QC. Tails pointing clockwise indicate the these simulations appear to be less unique under RMSE than they would be using the training QC. Perhaps surprisingly, the differences for the simpler model appear much more variable, but we also note that most of the larger discrepancies result in similar changes using the global and local evaluation, meaning the bias is mostly in the mean performance, and less so in the uniqueness metric. We considered the option of using the training QC flags for the evaluation period, however this would result in different models having very different evaluation periods.*

The updated figure and caption is included here.

[Figure]

Figure 1: RMSE values for the global (x-axis) and local (y-axis) model simulations. Columns show the three fluxes, the forst row shows data for S_lin, the second row for short_term243. The tails of each point show where the local and global RMSE values would be if the same QC flags were used for training and evaluating (the intersection of meteorological and flux QC flags). Tails pointing toward the zero in each axis indicate the model would have performed better using these QC flags. In other words, a tail pointing towards the origin means that our evaluation method has a bias toward worse mean RMSE, and a tail pointing clockwise from the origin indicates that our method has a bias towards lower uniqueness.

> Pg 7, line 17: Possibly using a combined summary statistic could simplify the procedure a bit? e.g. Gupta et al 2009

As noted in Gupta et al. (2009), their combined statistic still suffer from a pareto optimality problem,

in that there are multiple not-strictly-worse choices for weighting any combination. Additionally, such combined metrics reduce transparency about where problems are coming from. For these reasons, we felt it was more sensible to keep the metrics separate.

> Pg 7, line 24: Was there any attempt to prevent over-fitting, such a cross validation. How resistant is the cluster-plus-regression model to over-fitting?

Given enough data, the models are not sensitive to over-fitting. However, over-fitting on the local data is effectively part of the methodology. That is, the question is, how much does the functional form of the local data differ from the global data. That difference may be due to measurement error or physical site differences (these are not separable from a data perspective).

Of course, there is the possibility of noise in the local training data skewing the metric for individual sites. Our choice to use a minimum sample size of 5*n_vars per cluster for the cluster-and-regression models limits over-fitting to a large degree, while being small enough to not exclude all sites. On top of this, we are using multiple empirical models. These models should have minimal biases, and those biases should cancel when taking the multi-model mean. We are also looking for patterns across 100-155 sites in each analysis, using a GAM smoother, which also reduces the effects of individual model error on the patterns in the metrics substantially.

> Pg 8, lines 1-10: Again, a sensitivity analysis of how model and QC selection effect the metric would give users more confidence in the metrics.

We have included separate GAM interpolations, as noted in response to a comment above. We hope that this is satisfactory.

> Pg 8, line 8: While these caveats have been raise, really some effort to test the impacts of each should be included in the manuscript. I think it would be more beneficial to show the sensitivity of the metric rather than the extensive hypothesis testing.

This is a valid point, and we have moved a number of the results sections into the supplementary material, see the next point.

Of the caveats, the training testing mismatch is already shown in Figure 1, and the difference between metrics is shown in Figure 2. We can not see a way of testing the impact of the model failures, as complex models for those sites will not run successfully.

> Pg 9-11: While I appreciate the thorough analysis, I feel that the paper is lacking focus and comes across as sort of a data-dump. Especially given the fact that many of the results are inconclusive. Possibly focusing on some of the most promising hypothesis and moving many others to the supplemental material, or removing and simply mention that they were tested and the results were inconclusive. In all honesty I had to take a break from reading the paper after finishing this section.

This is a fair assessment. We have have moved a number of results sections into the supplementary material. The sections that we removed are:

- Aridity: partial duplication of the Mean Precip/Budyko sections, and less informative. We added a sentence to the Budyko Framework section to note the addition to supplementary.
- Interannual variability, Diurnal ranges, and Seasonality: These are kind of addenda to the Mean Temp/Precip section, and are not very informative. We added a sentence noting their presence in the supplementary material to the Mean Temp/Precip paragraph.

- Remoteness: This is already somewhat evident in the map figure. We have left the text alone, other than to note the presence of this figure in the supplementary material.

  Pg 12, line 5: Is there a reason for using the CRU mean annual temperature and not the mean annual temperature from the sites themselves?

The reason is that the CRU data spans a longer period, and is therefore more representative as a site characteristic MAP/MAT. Some sites have short record lengths and variable climates and so a years MAT/MAP (noting likely data gaps), would not be instructive. This was already partially explained at p.8, l.20 in the original submission, but we have modified the sentence to read:

*For example, if the site only has a short dataset measured over a particularly wet or dry period, or if a site has a strong seasonal pattern in the quality of the temperature data, the mean would be less representative of the general site characteristics than a longer-term dataset.*

  Pg 12, line 29: Seems an editing note got left in "(QUANTIFY?)". It would probably be a good idea to quantify what you mean by cooler.

Whoops! We changed this to "(< -5°C)".

  Pg 13, Figure 2: There seems to be some anti-correlation between the uniqueness and mean metrics, particularly with the RMSE of NEE. Is this likely just spurious?

Part of the reason the RMSE mean is lower at lower sites is that these sites are less energetic, and there is less variance in the fluxes, so modelling errors are correspondingly larger. This is part of the reason that we chose to stick mostly with the RMSE uniqueness for later figures in the paper, as it is less confounded, as well as being more immediately understandable compared to the other two metrics.

  Pg 14, line 4: Typo: "a possible a lower", furthermore, it would be beneficial to be more exact, as the results often report that there may be patters.

Thank you, fixed.

We have changed the text in the paragraph describing the Figure 2 to make it clearer that by "patterns" we mean patterns in over-all behaviour of the metrics contingent on the predictors:

*In Figure 2, we see that there are some consistent behaviour in the predictability metrics, which might indicate that mean annual temperature is a driver of predictability, but in general any consistency in the behavioural patterns are not strong. For instance, for RMSE uniqueness (first row), we see a slight increase in uniqueness (or lack of predictability) in sites that are cooler (< -5°C), as well as sites that have a mean annual temperature around 20°C, for both NEE and Qle. That pattern is less distinct in Qh. There is a stronger trend in RMSE mean (second row) for Qle, but this is likely largely due to the fact that warmer sites naturally tend to have larger heat fluxes. It seems surprising that Qh does not exhibit the same behaviour, since it is more directly related to temperature. Correlation uniqueness (third row) and mean (4th row) shows a similar pattern to RMSE uniqueness for NEE and Qle, where cooler sites and sites around 20°C tend to be harder to predict well. Patterns in Overlap uniqueness (5th row) and mean (6th row) are less clear, but there may be a slight indication of higher uniqueness around 20°C for NEE, and possibly a lower distribution predictability at higher average temperatures. Note that the negative Overlap uniqueness values are largely due to the fact that regression models do not perform particularly well on extreme values (as indicated in Best et al., (2015). We note that the two subset GAM plots for linear and*

*longer-term models describe similar behaviour in each metric in most panels, here and in later plots. The main differences seem to be largely to do with the more complex models' ability to capture more of the variance: The mean performance of these models under each metric is better (and the linear models' worse) than the mean, and the uniqueness is higher for the Corr and Overlap metrics, but quite similar for RMSE.*

> Pg 15, Figure 4: The use of two colormaps with overlapping colors can be confusing, giving a false indication that the yellow in both plots is related.

The use of separate colour schemes is simply to indicate that the plots are showing different variables. There are only a limited number of potential colour schemes available when aiming to maximise accessibility. In both cases, darker is higher, so we don't think that this is a real problem.

> Pg 16, line 9: Again, using CRU for precipitation data when you have site level data seems curious.

See reply to the comment on Pg 12, line 5.

> Pg 16, line 18: Is high diurnal temperature range not related to mean temperature? Can you differentiate this signal from that seen in Figure 2?

Actually, higher diurnal temperature range is more directly correlated with a lower mean precipitation/humidity. Since we have moved this figure to supplementary, and it does not form a core part of the paper, we have simply added a note to this effect in the figure caption, and hope that this suffices.

> Pg 17, Figure 6: Deviance from Budyko curve is never explicitly defined.

We added an extra sentence to the text paragraph describing the figure, and modified the following sentence:

*We also calculate a "Budyko deviance", which is simply the difference between the actual and predicted values on the Budyko plot, normalised by the predicted values, such that sites falling further above the Budyko curve have a positive deviance. There does not appear to be any pattern in predictability for NEE or Qle as a function of deviance from the Budyko curve (see second row in Figure 5), however there does seem to be some trend toward higher uniqueness for Qh for sites further above the Budyko curve.*

> Pg 18, line 2: "met forcing", met. is an abbreviation.

Replaced with "meteorological"

> Pg 18, line 9: "Shrubland and Savannah, and Grass", => Shrubland, Savannah, and Grass?

The "Grass" is the start of the next clause. We have replaced the comma with a semicolon:

*. . . more unique than all Forest types, Shrubland and Savannah; and Grass also tends to be more unique than . . .*

> Pg 22, line 2: parentheses has no close.

Fixed.

Pg 22, line 8: This is an example of a concrete example which give more confidence in the metric, yet it is given little attention compared to other analysis which are relatively inconclusive. If other examples exist possibly they could be highlighted.

There are other examples, although none that we can see as as obvious as the Metolius sites. We have extended the paragraph to read:

*There are interesting differences within clusters of FLUXNET sites, for example the US Metolius sites (US-Me1, US-Me2, US-Me6) are similarly unique for Qh and Qle, but US-Me1 is substantially more unique for NEE, and this site was measured for two years after a fire that killed all trees at the site (Law, 2016). This gives some indication that our uniqueness metric does indeed have bio-physical meaning. A similar though less distinct pattern can be seen in the CA-SF sites in Saskatchewan - the CA-SF3 site was burnt much more recently than the other two. There is also a notable gradation in Qle predictability in the UCI burn sites (CA-NS), that correlates with time since the last burn. There are likely other comparisons that can be drawn with sites not included in Tier 1, and an extended year-by-year analysis might also pick up land use changes related to cropping, for example.*

Pg 23, Figure 15: Could this figure be organized in a way that gives more information, such as ordering by uniqueness or grouping by PFT? In the current state it would maybe be more useful as a table with actual numbers.

The name-ordering provides a useful look-up table for sites, as well as grouping neighbouring sites in many cases. Ordering by uniqueness (of one variable? Of the average across variables?) would highlight the most and least unique-sites, but we are explicitly trying to avoid this, as discussed in response to the second comment.

We have added numbers to the plot, so it is now effectively a coloured table. In doing so, we encountered two bugs, which are now fixed, namely that the last site, ZM-Mon, had been dropped off the table due to a rounding error, and more seriously that each panel of the figure had been plotted with an independent colour scale.

The figure is now substantially different (in particular, the 2nd, 4th, 5th, and 6th panels are substantially lighter), although none of the text needed changing, since none of the comparison we made were affected by this change.

Pg, 24, line 5-7: I don't follow your logic here. I am not sure how the lack of a strong trend in Figure 12 provides support to the methodology. Also, I would not conflate the proximity of one tower to other towers with biome representativeness.

This is a fair point, we have removed the sentence.

---

## Author Comment (AC2) · 20 Jun 2018

> This manuscript presents a methodology for quantifying the "predictability" of land atmosphere fluxes of water, energy and carbon across 155 eddy covariance sites, with the goal of helping to better interpret comparisons between these observations and output from land surface models. This idea has considerable merit, and could be of interest to a large number of land surface model developers, and other synthesizers of eddy covariance data sets. Unfortunately, in its current form it is difficult to extract the most important information, as there is insufficient emphasis on what might be valuable, and much material is included which is not relevant. Overall, it needs to be much more focused, and the authors need to concentrate on: (i) predictability and their predictability metric; (ii) the models used, whilst greatly streamlining the hypotheses, given the inconclusiveness of the majority of the analysis.

We thank the reviewer for their insightful comments, and are pleased they see merit in our approach. We had attempted to answer each individual comment below, and we think that in doing so, we have made some substantial improvements to the paper.

We note in particular the general concern with the lack of clarity around the definitions of predictability and uniqueness. We have edited the manuscript in multiple places in an attempt to clarify our meanings here, and have separated the use of the words "predictability" and "uniqueness": "predictability" is only used for the general sense of over-all predictability, and "uniqueness" is only used to refer to the metric described in the paper.

We also note the reviewer's concern about the affect of using multiple different models, and we have added a number of clarifications and extra analyses to the paper, which are described below.

**General Comments**

1. As the authors acknowledge, "there is no single definition of predictability" – and this is a key challenge to this paper. The introduction needs to address this much earlier than the bottom of page 3, so there aren't several pages of text discussing something that has not been defined, or at least how it is being treated in the context of this manuscript.

We have changed this sentence to read:

*Predictability can broadly be defined as the ability to reproduce a property of a system, given only knowledge of variables that are causally related to that property.*

We think that that this is an all-encompassing definition of predictability. What is less defined is how to measure predictability - it is possible that there is no single metric that will cover all aspects of predictability. The following paragraph already covers some of those in an effort to illustrate that idea. We don't think that the broad concept of predictability, the meaning of which is fairly self-evident in the word itself, really needs to be spelled out more than this earlier in the introduction.

2. It seems that the authors are treating predictability as the inverse of "uniqueness", which is characterized as the deviation between a globally optimized model, versus

a locally optimized model. This needs additional clarification and justification. With this definition, the predictability is inherently model dependent, rather than some intrinsic property of the site alone. This is always the case possibly, but needs to be spelt out. It also highlights the importance of the models.

We are treating non-uniqueness as one aspect of predictability.

Predictability is an aspect of the relationship between flux datasets and meteorological datasets (and possibly other datasets). There is most likely no way to characterise the strength of that relationship other than by modelling it using data mining/statistical techniques. Information theory techniques might work conceptually, for example we could use something along the lines of mutual information for relationships between single met variables and single flux variables, but multivariate mutual information is a underdeveloped technique at best, and also does not capture transformations of variables (such as the impact of historical variable values) that are possible when modelling.

Yes, the metrics are somewhat model dependent, but the models we have used are structurally data agnostic, and the driving variables were chosen specifically because they add predictive value to simulations. In any case, predictability is necessarily a relative measure, and as long as the same procedure is used across all sites, it can be informative regardless of model choice. We also hope that the changes that we have made to Figure 1, and each of the GAM plot figures helps to alleviate concerns about model sensitivity potentially having a major impact on the qualitative results of the study.

3. A reliance on this uniqueness as a proxy for predictability seems like it might have drawbacks. Consider the NEE plots in Figure 1. There are a group of sites with relatively high "uniqueness", and thus low predictability, but with a global RMSE less than 7, which is lower than for a large fraction of sites indicating a better mean performance. Are these sites more or less predictable? It seems that this can only be quantified through combing the uniqueness metric with the mean performance into a single metric, but uniqueness is discussed in isolation throughout the results section.

Again, uniqueness is one aspect of predictability. On reading through the paper, it is clear that we were not always rigorous in our terminology. We have now been through the manuscript, and have made sure that every case where "predictability" is used, that it's actually in the general sense, and that it is always clear that uniqueness is simply a major component of predictability. We use "predictability metrics" to refer to all combinations of mean/uniqueness and RMSE/Corr/Overlap metrics.

4. This can be addressed by combining the metrics with appropriate weights. The authors say this is not done due to the difficulty in combining the different metrics, but given the lack of information in the metrics other than RMSE, and the apparent requirement to combine uniqueness and mean performance, this should be reconsidered, with at least these two components of each metric combined.

The problem with combining metrics is that you lose a substantial amount of nuance. This is particularly the case when one metric has a strong, but meaningless trend in it - for example RMSE over mean temperature (this is obviously not meaningless in the broad sense, but it is not useful for predictability assessment, except as something to be aware of). On top of this, a sum (weighted or

not) suffers from the problem that the mean metric is linear, while the uniqueness metric is radial, and so the uniqueness metric dominates at small RMSE values, and the mean dominates at large RMSE values.

In our results, there are very few cases where the mean performance metric has any clear patterns in it that are either not explainable in this way, or not already visible in the uniqueness metric. As such, while mean metric performance is important to take into account, uniqueness is the more important and interesting component of predictability, as far as comparing FLUXNET sites goes.

5. What is the additional information gained from switching from Cartesian to polar coordinates? Would not a simple mean of the global and local models, and the normalized difference between then suffice?

The benefit of using polar coordinates is that uniqueness is orthogonal to mean performance. Using the angle between the metrics is almost equivalent to using the 1 - log(local/global), except that the normalisation is problematic for metrics that often have values below 1. The angle method is the same for all metrics, normalisation would not be.

6. This manuscript relies heavily on previous work (Best et al, 2015, Haughton et al., 2018). Indeed, it is not possible to understand much about the models with out consulting this closely. Given how dependent the predictability metrics are on these models, some further description of them is required here.

The models are relatively simple, and there is a not a lot more to describe, but we have expanded the model description somewhat. This point is substantially similar to one of Reviewer 1's comments, and we copy that here:

The beginning of the 4th paragraph of the Methods already describes the cluster-plus-regression models - they are conceptually simple, and there is not a lot more to say, but we have split this paragraph in two, and changed the first part to add some description of the use of the cluster-plus-regression models:

*This procedure is model-agnostic, and we have used models in the framework developed in Best et al. (2015) and Haughton et al. (2018), because they are conceptually simple, but able to fit complex functional relationships. These models (listed in Table 1) include some simple linear regressions, as well as cluster-plus-regression models. The cluster-plus-regression models consist of a K-means clustering over meteorological driving data, and then an independent linear regression between drivers and fluxes at each cluster. These cluster-plus-regression models can fit arbitrary functional forms between predictor and response variables, when using a high enough cluster count (k), and given enough data. The models are not perfectly deterministic, since K-means convergence is dependent on cluster initialisations, but the variance in the results is small (see supplementary material, Haughton et al., 2018), and unlikely to substantially affect our results substantially. Our use of an ensemble of models at each site further mitigates this problem. The ensemble also allows us to overcoming the problems of the simpler models failing to capture behavioural nuances, and of the more complex models failing to train at some sites due to insufficient data (described below).*

We also moved the note about which fluxes are modelled to the top of the methods section.

We have added the long-form of each long-term/short-term model name to the table.

7. Indeed, it's very unclear why multiple models are being used at all? What is the benefit of doing this rather than using the single "best" model?

The "best" models - the most complex ones - often fail locally due to not having enough training data. On the other hand, the simple models clearly only capture a subset of the behaviours at any given site. By using an ensemble of models, we also minimise model-related variance. We have modified what is now the 3rd paragraph of the methods section to address these points:

*This procedure is model-agnostic, and we have used models in the framework developed in Best et al., 2015 and Haughton et al., 2018, because they are conceptually simple, but able to fit complex functional relationships. These models (listed in Table 1) include some simple linear regressions, as well as cluster-plus-regression models. The cluster-plus-regression models consist of a K-means clustering over meteorological driving data, and then an independent linear regression between drivers and fluxes at each cluster. These cluster-plus-regression models can fit arbitrary functional forms between predictor and response variables, when using a high enough cluster count (k), and given enough data. The models are not perfectly deterministic, since K-means convergence is dependent on cluster initialisations, but the variance in the results is small (see supplementary material, Haughton et al., 2018), and unlikely to substantially affect our results substantially. Our use of an ensemble of models at each site further mitigates this problem. The ensemble also allows us to overcoming the problems of the simpler models failing to capture behavioural nuances, and of the more complex models failing to train at some sites due to insufficient data (described below).*

8. Although the authors suggest they want to leave this for future work as it is "substantially more complex", it seems at least some examples are required to explain how the predictability metrics are sensitive to the models, and how this can be interpreted when discussing specific sites. For example, it seems that many semi-arid sites are characterized by these models to have high uniqueness, but what would happen if soil moisture was included in the model?

We have updated Figure 1 to now include short_term243, in order to show how much the change in QC affects a more complex model. The paragraph describing it and the figure and caption have been modified to read:

*The uniqueness and mean performance metrics are shown for RMSE in Figure 1 for the S_lin and short_term243 models to illustrate how to interpret later figures:* uniqueness *is the angle measured clockwise from the origin (the optimal metric value) and the 1:1 line (equal local and global performance), and* mean performance *is the average performance of the local and global simulations, given by the distance of each point from the origin. Each point is a different site. Figure 1 also illustrates the differences between the results when the local training data is identical to the testing data, and when it differs due to mismatch between the meteorological and flux QC flags between training and testing. In each panel, the blue points indicate the local and global RMSE values used for the simulation in the remainder of the study. The tail from each point indicates where these values would have been if the same QC data that was used for training was used for evaluation (meteorological + flux QC, instead of just Flux QC. The tail points are strictly at or below the 1:1 line (as the empirical fit is optimised for RMSE locally, but not globally). The flux-only QC evaluated blue points can shift, and some lie very slightly above the 1:1 line. Tails pointing towards the origin indicate that these simulations' mean RMSE is worse than it would be using the training QC. Tails pointing clockwise indicate the these simulations appear to be less unique under RMSE than they would be using the training QC. Perhaps surprisingly, the differences for the simpler model appear much more variable, but we also note that most of the larger discrepancies result in similar changes using the global and local evaluation, meaning the bias is mostly in the mean performance, and less so in the uniqueness metric. We considered the option of using the training QC flags for*

*the evaluation period, however this would result in different models having very different evaluation periods.*

[Figure]

Figure 1: RMSE values for the global (x-axis) and local (y-axis) model simulations. Columns show the three fluxes, the forst row shows data for S_lin, the second row for short_term243. The tails of each point show where the local and global RMSE values would be if the same QC flags were used for training and evaluating (the intersection of meteorological and flux QC flags). Tails pointing toward the zero in each axis indicate the model would have performed better using these QC flags. In other words, a tail pointing towards the origin means that our evaluation method has a bias toward worse mean RMSE, and a tail pointing clockwise from the origin indicates that our method has a bias towards lower uniqueness.

We have also updates all of the GAM plots (Figures 2, 3, 5, 8, 9, 10) now include separate GAM fits for the longer-term models and the linear models. We added the following text to the paragraph describing Figure 2:

*We note that the two subset GAM plots for linear and longer-term models describe a similar pattern in each metric in most panels, here and in later plots. The main differences seem to be largely to do with the more complex models' ability to capture more of the variance: the mean performance of these models under each metric is better (and the linear models' worse) than the mean, and the uniqueness is higher for the Corr and Overlap metrics, but quite similar for RMSE.*

Unfortunately soil moisture is not reliably available in the FLUXNET dataset, which is a large part of the reason this paper and Haughton et al (2018) did not include it. The humidity and rainfall lags in the long_term243 model act as a proxy for soil moisture, however, the long_term243 model

did not run successfully locally at enough sites to provide an adequate picture of predictability contingent on those proxies.

9. Whilst uniqueness as defined here is certainly a useful metric to assess flux sites by, and to help interpret comparisons between observations and land surface model output, it is unclear it represents something like the inverse of "predictability". In fact, a contrary argument could be made that the sites that exhibit large reductions in model error when optimized with local data are the most predictable. Whilst for those sites that don't see model improvement when just local data are used this lack of sensitivity might also be interpreted as a lack of predictability, particularly for sites with low mean performance. In this context, a predictable site is one where given more information, model skill increases, and whilst at an unpredictable site specific information does not increase skill.

This was a very thought-provoking comment, and provided us with the impetus to try to set out the relationship between predictability and uniqueness and mean performance more clearly.

We have added the following discussion of the relationship between uniqueness and predictability and corresponding figure to the beginning of the discussion:

*In our exploration of meteorological predictability, which we characterised using both uniqueness and mean performance for each metric (described in the Methods), we have mostly focussed on uniqueness of behaviour of sites. This metric is the most novel component of this study, and is, we think, the most interesting aspect of relative site predictability. However, it is also less intuitive than mean performance. It is worth reiterating that uniqueness is* not *the direct inverse of predictability, and under certain conditions it can actually be correlated with overall predictability. In an effort to clarify uniqueness in an intuitive way, we illustrate the relationships between meteorological drivers and fluxes, and site-specific (non-meteorological) modulators of these relationships in Figure 11.*

*Figure 11 provides a schematic for understanding how mean performance and uniqueness interact as components of predictability. In all cases, we assume that the observations include some noise, or unpredictable components of variability. In the left column, we present a case where there is a strong universal relationship between meteorological forcings and flux observations (in this case for example, a linear trend), in the right column, this relationship is weak or non-existent. In the top row, we have the case where the local site conditions modulates the behaviour of the fluxes (in this case for example, a simple bias, but it could also be a complex non-linear relationship), and in the bottom row, there is no such site-related modulation. In the right column, where there is no meteorological driver relationship, we can see that uniqueness is correlated with performance: in the absence of site modulators (bottom right) there is no predictability (as all variability is due to noise) and both models perform poorly. In the presence of site modulators (top-right), predictability is higher and uniqueness is also higher (because the local model performs better). When there are strong meteorological drivers (left column), predictability is higher in the case where site modulators are weak (bottom left) because the global model is able to perform well. In this case, sites with strong site modulators are less predictable because the global performance is worse, and uniqueness would clearly be anti-correlated with "predictability". It is clear that there are strong relationships between meteorological drivers and fluxes (see Best et al., 2015, Haughton et al., 2018), and so the inferences made in this study mostly fall in the left column. This suggests that in real-world situations uniqueness is mostly anti-correlated with predictability, but not perfectly, and so we emphasise that uniqueness alone is not an adequate proxy for predictability.*

[Figure]

Figure 2: Schematic of sources of variability and how they affect predictability, mean performance, and uniqueness. The black line represent the flux observations, and the grey ribbon, the unpredictable internal variability or noise in the system. "Meteorological drivers" indicate universal physical relationships between meteorological forcings and fluxes, here we have indicated this using a simple linear trend as an example (red and blue lines). "Site modulators" are characteristics of the site (soil or vegetation properties, storage pools, geography, or data problems) that modulate the meteorological -> flux relationship such that it is different to the relationships observed globally - here we have used a simple bias as an example.

10. Both sections 2.2 and 3.2 read as an overly long laundry list of "everything we tried". It will be easy to greatly increase the overall focus of the manuscript by addressing this. Given the lack of conclusiveness regarding the majority of the hypotheses about determinants of predictability, a brief note that they were considered and findings were inconclusive is all that is required.

We have significantly reduced the information in both of these sections. See our response to Reviewer 1's comment on Pg 9-11.

**Specific Comments**

P1 L1-8. Rather like the manuscript as a whole, the abstract needs much more focus. Emphasis specific detail, not background information and motivation.

We have re-written the abstract to include less background and more results. It now reads:

*The FLUXNET dataset contains eddy covariance measurements from across the globe, and represents an invaluable estimate of the fluxes of energy, water and carbon between the land surface and the atmosphere. While there is an expectation that the broad range of site characteristics in FLUXNET result in a diversity of flux behaviour, there has been little exploration of how predictable site behaviour is across the network. Here, 155 datasets from the Tier 1 of FLUXNET2015 were analysed in a first attempt to assess individual site predictability. We defined site **uniqueness** as the disparity in performance between multiple empirical models trained globally and locally for each site, and used this along with the mean performance as measures of predictability. We then tested how strongly uniqueness was determined by various site characteristics, including climatology, vegetation type, and data quality. The strongest determinant of predictability appeared to be that drier sites tended to be more unique. We found very few other clear predictors of uniqueness across different sites, and in particular found little evidence that flux behaviour is well discretised by vegetation type. Data length and quality also appeared to have little impact on uniqueness. While this result might relate to our definition of uniqueness, we argue that our approach provides a basis for site selection in LSM evaluation, and invite critique and development of the methodology.*

P4 L13-21. Not methods.

We have moved this section into the introduction, and have adjusted the surrounding text to suit this change.

P8 Fig 1. Don't understand the need for colored dots?

As stated in the caption, "Colours simply serve to identify sites, and allow clearer comparison between the top and bottom rows".

P12 L29. Yes, you might want to [QUANTIFY] that

Whoops! We changed this to ($< $ -5°C).

P15 Fig 4. Presumably it is the mean values that are being plotted here?

Yes, we added a parenthetical remark to the figure caption to note this. The first line of the caption now reads:

*Predictability metrics for mean annual temperature vs mean annual precipitation (mean across models).*

P15 Fig 4. Seems like NEE needs a different scale?

Yes. Since the relative differences are of interest, not the absolute values, instead of giving NEE a different scale, we have multiplied the NEE mean RMSE by 10, and added a note to that effect to the caption.

P16 Fig 5. The two sites with an aridity index higher than 3 haven't been excluded.

This was left over from a previous version of the figure. In any case, we have removed Aridity from the paper and put it in the supplementary material, and have adjusted the caption to suit.